# Deep Extended Hazard Models for Survival Analysis

**Qixian Zhong**
Department of Statistics and Data Science
School of Economics and Wang Yanan Institute for Studies in Economics (WISE)
Xiamen University
qxzhong@xmu.edu.cn

**Jonas Mueller**
Amazon Web Services
jonasmue@amazon.com

**Jane-Ling Wang**
Department of Statistics
UC Davis
janelwang@ucdavis.edu

## Abstract

Unlike standard prediction tasks, survival analysis requires modeling right censored data, which must be treated with care. While deep neural networks excel in traditional supervised learning, it remains unclear how to best utilize these models in survival analysis. A key question asks which data-generating assumptions of traditional survival models should be retained and which should be made more flexible via the function-approximating capabilities of neural networks. Rather than estimating the survival function targeted by most existing methods, we introduce a Deep Extended Hazard (DeepEH) model to provide a flexible and general framework for deep survival analysis. The extended hazard model includes the conventional Cox proportional hazards and accelerated failure time models as special cases, so DeepEH subsumes the popular Deep Cox proportional hazard (DeepSurv) and Deep Accelerated Failure Time (DeepAFT) models. We additionally provide theoretical support for the proposed DeepEH model by establishing consistency and convergence rate of the survival function estimator, which underscore the attractive feature that deep learning is able to detect low-dimensional structure of data in high-dimensional space. Numerical experiments also provide evidence that the proposed methods outperform existing statistical and deep learning approaches to survival analysis.

## 1   Introduction

Across areas such as biomedical science and reliability engineering, survival data analysis is critically used to study the time until certain events occur (e.g. patient death in clinical applications, component failure in industrial applications). A key issue stems from the fact that many occurrence-times fail to be recorded due to natural data censoring. For example, a patient may relocate and drop out of a longitudinal clinical study, leading to the loss of follow-up observations for the remaining period of the study. For such right censored data, we only know that the true event time is greater than the last available observation time for this individual (but not by how much), and thus standard regression approaches are not applicable to most survival data. A naive approach that removes all censored individuals from the data would induce bias, as these tend to be individuals with longer event times.

35th Conference on Neural Information Processing Systems (NeurIPS 2021).

Learning the distribution of the event time in the presence of censorship is a fundamental task in survival analysis. A wide array of statistical methodologies have been developed to study this distribution. Among them, Kaplan and Meier [37] considered a product of successive conditional probabilities to estimate the *survival function*, which encodes tail probabilities of the event time distribution. As a nonparametric approach, the Kaplan-Meier estimator assumes no particular form for either the underlying distribution of event time or the relationship between individuals' observed feature-values and the outcome event time. Alternatively, the Nelson-Aalen estimator considers the cumulative hazard function from a counting process perspective [1, 48]. The classical Nelson-Aalen estimator was improved by leveraging more powerful supervised learning methods, in particular random forests, which lead to the *random survival forest* model [33].

While the Kaplan-Meier estimator has been extended to assess the effect of multiple subject features (covariates) on the associated event time [5], it struggles to work with even moderately high-dimensional features. A semi-parametric approach, the Cox proportional hazards (CoxPH) model [14], has thus been popularly adopted as a dimension reduction approach. However, the CoxPH model is restrictive as it assumes that the conditional log hazard functions for individuals are parallel to each other. When this assumption fails to be realistic, the accelerated failure time (AFT) model [35, 62] often emerges as an alternative, which simply assumes a linear regression model for the log event time against the individuals' feature-values. Like the CoxPH model, the AFT model is also semi-parametric because the error distribution in the regression model is unspecified. An even simpler approach is to adopt a parametric approach for survival data by assuming that the distribution of event time obeys a certain known law, such as the Weibull or Gamma distribution [44]. While non-parametric methods are more flexible, a semi-parametric or parametric approach can be more effective when there are limited data, if the distribution of event times can be closely approximated by a certain semi-parametric or parametric form.

Rapid advances in data collection and analysis have encouraged researchers to explore the potential of deep learning in the field of survival analysis. Faraggi and Simon [20] proposed to replace the linear predictor of the CoxPH model with a single hidden-layer neural network, but this extension failed to provide reliable improvements in terms of concordance index [63]. Under the same CoxPH framework, Katzman et al. [38] and Kvamme et al. [41] applied neural networks of greater complexity to obtain more flexible nonlinear models. Zhu et al. [67] and Tarkhan et al. [57] used convolutional architectures to explore the relationship between event time and unstructured features like images, and domain-specific variants of these models have been developed for particular applications, such as genomic data [10, 26], clinical research [49], pedestrian waiting time in urban areas [34] and heart failure rehospitalization [36].

Beyond the CoxPH framework, deep learning has been used to model survival data in various alternative forms, including: neural network versions of the Beta-Logistic model [32], recurrent neural networks applied with time-discretization [22, 51], generative adversarial modeling of time-to-event distributions [8], generalized forms of regression [40, 43], and more flexible extensions of the Kaplan-Meier estimator [9, 56] and the Weibull distribution [50]. Among existing deep learning methods for survival analysis, the distribution free approaches [9, 12, 28, 40, 43] make almost no assumptions on the underlying survival function structure, but these very general models may exhibit poor sample complexity, requiring very large data to accurately estimate the survival function.

In this paper, we adapt neural networks to generalize the extended hazard (EH) model [11], which includes both the CoxPH and AFT models as special cases. The proportional hazards assumption are often violated in practice and can lead to misinterpretation of the results, especially when there is heterogeneity among individuals [2]. In contrast, the AFT framework directly considers the relationship between feature values and the event time, which results in a time-dependent hazard ratio that is more realistic. Keiding et al. [39] also show that, relative to CoxPH, AFT models are more stable when accounting for unobserved features. Other studies of CoxPH and AFT models have however reported conflicting results [46], and thus practitioners may find it difficult to choose which model to use in their applications. The EH model [19, 58, 59], under which feature values can affect both the baseline hazard rate and hazard ratio, flexibly combines the CoxPH and AFT models into a more general framework. EH models also subsume a wider range of survival models such as Weibull and Gamma distributions. Compared with the current practice of deep learning under the CoxPH or AFT framework, our deep extensions of EH model retain the advantages of standard EH model over CoxPH and AFT, and achieve better empirical performance on real survival data.

While deep learning has enjoyed a number of breakthroughs in survival data analysis, mathematical understanding of its success is still lagging far behind. Leveraging neural networks' capacity for nonparametric function approximation [4, 17, 29, 53, 65], this paper provides theoretical support for our Deep Extended Hazard (DeepEH) model, which replaces the two linear risk predictors in the standard EH model with two nonparametric functions (cf. $h_1$ and $h_2$ in (4)). Using neural networks to model these nonparametric functions, we show that the resulting estimators of $h_1$, $h_2$ and survival functions are asymptotically consistent and enjoy fast convergence rates. Specially, the convergence rates are determined by the intrinsic dimension of the underlying functions rather than the original high-dimension input features. Our results formally illustrate how neural networks are able to identify low-dimensional structure of the data, hence circumventing the curse of dimensionality.

## 2 Background

In survival analysis, we are interested in modeling the time $T$ until some event of interest (i.e. survival time), whose probability distribution is typically characterized in one of four equivalent ways: probability density function, survival function, hazard function, or cumulative hazard function. The *survival* function $S(t)$ is the probability of a individual surviving beyond time $t$:

$$S(t) = \mathbb{P}(T > t) = 1 - \int_0^t f(s)ds,$$

where $f$ is the probability density function of survival times. The *hazard* function, denoted by $\lambda(t)$, characterizes the chance that an event occurs in an infinitesimal interval after time $t$, given it has not yet occurred at time $t$:

$$\lambda(t) = \lim_{\delta \to 0} \frac{\mathbb{P}(t \le T < t + \delta | T \ge t)}{\delta}.$$

This results in the relationship: $S(t) = \exp\{-\Lambda(t)\}$, where $\Lambda(t) = \int_0^t \lambda(s)ds$ is the *cumulative hazard* function.

In practice, a survival study of $n$ subjects will contain less than $n$ completely observed event times due to right censoring. For instance, the event of some individuals may not have occurred yet at the time when the study is concluded or the subject may have dropped out of the study. For the $i$-th subject in the study, $T_i$ and $C_i$ denote, respectively, the event time and the potential censoring time; and $X_i \in \mathbb{R}^p$ denotes the observed features (covariates). Thus, the observed data from a typical survival study contain independent observations $\mathcal{D} = \{X_i, O_i, \Delta_i\}_{i=1}^n$, where the observed time $O_i = \min(T_i, C_i)$ and the event indicator $\Delta_i = 1$, if the observed time is $T_i$, i.e. $T_i \le C_i$, otherwise $\Delta_i = 0$, the subject is censored at $C_i$. Given the feature values $X_i$, the event time $T_i$ and censoring time $C_i$ are typically assumed to be independent. We aim to estimate the survival function and hazard function using censored data in which the true event times may be unavailable. Conditional on the features $X$, the overall log-likelihood of the censored survival data $\mathcal{D}$ [15] is:

$$\mathcal{L}_n = \frac{1}{n} \sum_{i=1}^n \left[ \Delta_i \log f(O_i|X_i) + (1 - \Delta_i) \log S(O_i|X_i) \right] = \frac{1}{n} \sum_{i=1}^n \left[ \Delta_i \log \lambda(O_i|X_i) - \Lambda(O_i|X_i) \right],$$
(1)

where $f(t|X)$, $S(t|X)$, $\lambda(t|X)$ and $\Lambda(t|X)$ denote the conditional density, survival, hazard, and cumulative hazard function of $T|X$, respectively, and the last equality is derived from the aforementioned relationship between the (cumulative) hazard function and the density/survival function. To obtain maximum likelihood estimates, one can assume that the conditional distribution of $T$ follows a particular parametric distribution. However to avoid restrictive assumptions, it is advantageous to use more flexible alternatives such as the CoxPH and AFT models.

### 2.1 Cox Proportional Hazard Model

Given features $X$, the Cox proportional hazard (CoxPH) model [13] for event time $T$, is a semi-parametric model whose hazard function has a multiplicative form in terms of a baseline hazard function $\lambda_0(t)$ and an exponential risk function of $X$:

$$\lambda(t|X) = \lambda_0(t) \exp(\beta^\top X),$$

where the baseline hazard function $\lambda_0(t)$ and parameter $\beta$ are both unknown. The function $\lambda_0(t)$ represents the baseline hazard rate when all features take the baseline value of zero. It is difficult to estimate $\lambda_0$ and $\beta$ directly from the likelihood function (1) because the optimization with respect to $\lambda_0$, a nonparametric function, would be challenging. In fact, this likelihood is unbounded due to the unconstrained nature of the nonparametric function $\lambda_0$. Luckily the partial likelihood method [13, 14] has been developed to handle the parametric component of the model without involving the nonparametric component $\lambda_0(t)$. Specifically, the partial likelihood technique first estimates $\beta$ by maximizing the log-partial likelihood:

$$\mathcal{L}_n(\beta) = \frac{1}{n} \sum_{i=1}^{n} \Delta_i [\beta^\top X_i - \log(\sum_{j=1}^{n} Y_j(O_i) e^{\beta^\top X_j})],$$

where $Y_j(t) = 1(O_j \geq t)$. Intriguingly, the log-partial likelihood does not involve the nonparametric baseline hazard function $\lambda_0$, which is a major advantage in accomplishing dimension reduction. Once the parameter $\beta$ has been estimated through the log partial likelihood, the cumulative baseline hazard function $\Lambda_0(t) = \int_0^t \lambda_0(s) ds$ can be estimated through the Breslow estimator [6]. Here the simple parameter $\beta_0$ can be estimated directly through the partial likelihood without requiring nonparametric estimation of $\lambda_0$. Thanks to the appeal of this straightforward estimation, the Cox proportional hazards model has dominated the field of survival analysis for nearly fifty years.

## 2.2 Accelerated Failure Time

The accelerated failure time (AFT) model [62] is an alternative to CoxPH when the proportional hazards (PH) assumption is violated. Given feature values $X$, the hazard function for an AFT model is of the form:

$$\lambda(t|X) = \lambda_0(te^{\beta^\top X})e^{\beta^\top X}, \tag{2}$$

where $\lambda_0$ is an unknown baseline hazard function and $\beta$ is an unknown parameter. Unlike the CoxPH, the AFT model allows the feature $X$ to influence the time curve of the baseline hazard function as well. AFT with $\beta > 0$ assumes that a subject with a larger $X$ value will have double jeopardy through a higher risk factor $e^{\beta^\top X}$ and a faster hazard schedule $\lambda_0(te^{\beta^\top X})$ compared to the baseline hazard $\lambda_0(t)$, hence the name "accelerated failure".

## 2.3 Extended Hazard Model

In practice, it may be unclear whether a CoxPH or AFT model is more suitable for the data in hand. The Extended Hazard (EH) model [11] includes both the CoxPH and AFT model as special cases, thereby providing a more flexible and larger class of models to use in such practical applications. While AFT (2) allows for only one feature effect $\beta$, the EH model extends this is to allow two different feature effects $\beta_1$ and $\beta_2$:

$$\lambda(t|X) = \lambda_0(te^{\beta_1^\top X})e^{\beta_2^\top X}. \tag{3}$$

Note that the EH model collapses to the CoxPH model when $\beta_1 = 0$, and the AFT model when $\beta_1 = \beta_2$. The EH model can thus flexibly leverage the complementary strengths of the CoxPH and AFT model, adaptively exploiting whichever assumptions are more appropriate for a given dataset.

# 3 Methods

## 3.1 Deep Extended Hazard Model

Our DeepEH model is derived by extending the standard EH model in (3) to the following hazard function:

$$\lambda(t|X) = \lambda_0(te^{h_1(X)})e^{h_2(X)} \tag{4}$$

where $h_1(X)$ and $h_2(X)$ are unknown functions and $\lambda_0$ is an unknown and non-negative baseline hazard function. When $h_1 = 0$, it collapses to the popular DeepSurv [38]. When $h_1 = h_2$, we call this the Deep Accelerated Failure Time (DeepAFT) model as it is a deep version of the AFT model (2) that has been previously unexplored. All of our theory and methodology for DeepEH extends to DeepAFT as well, and we also present empirical results for DeepAFT in this work. Let

$\Lambda_0(t) = \int_0^t \lambda_0(s)ds$ be the cumulative baseline hazard function. By the definition of conditional survival function $S(t|X) = \exp\{-\int_0^t \lambda(s|X)ds\}$, the survival function of DeepEH is expressed as

$$S(t|X) = \exp\{-\int_0^t \lambda_0(se^{h_1(X)})e^{h_2(X)}ds\} = \exp\{-\Lambda_0(te^{h_1(X)})e^{h_2(X)-h_1(X)}\}. \quad (5)$$

Since the log-likelihood is unbounded, direct maximization of the likelihood function will not work. Instead, a pseudo likelihood may be employed instead Tseng and Shu [59]. Specifically, we first use a kernel smoothing method to approximate the hazard function with fixed $h_1$ and $h_2$ and then plug the the smoothed hazard function into the log-likelihood function (1). This yields the following pseudo-likelihood function for our DeepEH model:

$$\mathcal{L}_n(h_1, h_2) = \frac{1}{n}\sum_{i=1}^{n}\Delta_i[h_2(X_i) - R_i(h_1)] + \frac{1}{n}\sum_{i=1}^{n}\Delta_i \log\left[\frac{1}{nb}\sum_{j=1}^{n}\Delta_j K\left(\frac{R_j(h_1) - R_i(h_1)}{b}\right)\right]$$
$$- \frac{1}{n}\sum_{i=1}^{n}\Delta_i \log\left[\frac{1}{n}\sum_{j=1}^{n}\frac{e^{h_2(X_j)}}{e^{h_1(X_j)}}\Phi\left(\frac{R_j(h_1) - R_i(h_1)}{b}\right)\right],$$

where $K(\cdot)$ is a known kernel function satisfying Assumption (A4) below and we adopt the Gaussian kernel in our implementation, $R_i(h_1) = \log(O_i e^{h_1(X_i)})$, $b$ is a positive bandwidth constant, and $\Phi(t) = \int_{-\infty}^t K(s)ds$.

We use two neural networks to approximate the unknown functions $h_1(X)$ and $h_2(X)$ in (4). More specifically, we learn:

$$(\hat{h}_1, \hat{h}_2) = \arg\min_{g_1, g_2 \in \mathcal{G}}\{-\mathcal{L}_n(g_1, g_2) + \rho J(g_1, g_2)\} \quad (6)$$

where $\mathcal{G}$ again denotes a certain family of neural networks and $J$ is a regularization term weighted by $\rho \geq 0$ and it is set to be a $L^2$ regularization for the weights and biases of $g_1$ and $g_2$ in this paper. We estimate respectively the baseline hazard and cumulative baseline hazard functions via:

$$\hat{\lambda}_0(t) = \frac{\frac{1}{nt}\sum_{i=1}^n \Delta_i K_b(R_i(\hat{h}_1) - \log t)}{\frac{1}{n}\sum_{i=1}^n \frac{e^{\hat{h}_2(X_i)}}{e^{\hat{h}_1(X_i)}}\int_{-\infty}^{R_i(\hat{h}_1)-\log t} K_b(s)ds} \quad \text{and} \quad \hat{\Lambda}_0(t) = \int_0^t \hat{\lambda}_0(s)ds,$$

where $K_b(\cdot) = K(\cdot/b)/b$. Our implementation employs a Riemann sum of the estimated baseline hazard function $\hat{\lambda}_0$ to obtain $\hat{\Lambda}_0$, which leads to an estimate of the survival function based on (5):

$$\hat{S}(t|X) = \exp\{-\hat{\Lambda}_0(te^{\hat{h}_1(X)})e^{\hat{h}_2(X)-\hat{h}_1(X)}\}. \quad (7)$$

## 3.2 Theoretical Analysis

To study the asymptotic properties of our estimators, we assume the data follow the survival model specified in (4), and that our multilayer feedforward neural networks obey a special structure with parameters $\boldsymbol{d}, s, M$ and $A$ described below. Here a $M$-layer neural network with parameter $\boldsymbol{d} = (d_0, \cdots, d_{M+1})$ is a composition of functions:
$$g(x) = W_M g_{M-1}(x) + \mu_M, g_{M-1}(x) = \sigma(W_{M-1}g_{M-2}(x) + \mu_{M-1}), \cdots, g_0(x) = \sigma(W_0 x + \mu_0)$$
with activation function $\sigma$, weight matrices $W_k \in \mathbb{R}^{d_{k+1}\times d_k}$ and bias vectors $\mu_k \in \mathbb{R}^{d_k}$, for $k = 0, \cdots, M$. We consider a particular family of neural networks

$$\mathcal{G}(M, \boldsymbol{d}) = \{g : g \text{ is an } M\text{-layer network with parameters } \boldsymbol{d} \text{ and } \max_{k=0,\cdots,M}\|W_k\|_\infty \vee \|\mu_k\|_\infty \leq 1\},$$

where $a \vee b = \max(a, b)$ and $\|\cdot\|_\infty$ denotes the supremum norm of a matrix/vector. Neural networks used in practice often contain many parameters, which can lead to overfitting. To address this, Srivastava et al. [55] proposed dropout regularization, which randomly sets some of network's activations to zero during training. Inspired by dropout, we further consider another neural network family

$$\mathcal{G}(\boldsymbol{d}, s, M, A) = \{g \in \mathcal{G}(M, \boldsymbol{d}) : \sum_{k=1}^{M}\|W_k\|_0 + \|\mu_k\|_0 \leq s, \text{ and } \|g\|_\infty \leq A\}, \quad (8)$$

where $\|\cdot\|_0$ is the number of non-zero entries of matrix/vector and $\|g\|_\infty$ is the uniform norm of function $g$.

Our theory also assumes the underlying nonparametric estimands belong to the Hölder class of smooth functions with parameters $\alpha, B > 0$ and domain $\mathbb{D} \subset \mathbb{R}^p$ defined as:

$$\mathcal{H}_p^\alpha(\mathbb{D}, B) = \{h : \mathbb{D} \to \mathbb{R} : \sum_{\beta:|\beta|<\alpha} \|\partial^\beta h\|_\infty + \sum_{\beta:|\beta|=\lfloor\alpha\rfloor} \sup_{x,y\in\mathbb{D},x\neq y} \frac{|\partial^\beta h(x) - \partial^\beta h(y)|}{\|x-y\|_\infty^{\alpha-\lfloor\alpha\rfloor}} \leq B\},$$

where $\lfloor\alpha\rfloor$ is the largest integer strictly smaller than $\alpha$, $\partial^\beta := \partial^{\beta_1}\ldots\partial^{\beta_p}$ if $\beta = (\beta_1, \cdots, \beta_p)$, and $|\beta| = \sum_{k=1}^p \beta_k$. Many common functions satisfy the Hölder property, including: polynomials, trigonometric functions, exponential and logarithmic functions, as well as Lipschitz continuous functions when $\alpha = 1$. For some $L \geq 0$ and $\boldsymbol{p} = (p_0, \ldots, p_{L+1}) \in \mathbb{N}^{L+2}$, we also consider a composite Hölder function:

$$h = h_L \circ h_{L-1} \circ \cdots \circ h_1 \circ h_0$$

where $h_l : [a_l, b_l]^{p_l} \to [a_{l+1}, b_{l+1}]^{p_{l+1}}, l = 0, \cdots, L$. Denote $h_l = (h_{l1}, \cdots, h_{lp_{l+1}})^T$ and let $p_{lj}$ be the unique number of features that each $h_{lj}$ depends on. Define $p_l^* = p_{l1} \vee \cdots \vee p_{lp_{l+1}}$. For illustration, if

$$h(x_1, \ldots, x_9) = h_{11}(h_{01}(x_1, x_2, x_3), h_{02}(x_4, x_5, x_6), h_{03}(x_7, x_8, x_9)), \tag{9}$$

we have $p_0^* = p_1^* = 3$. Further, we assume each $h_{lj} \in \mathcal{H}_{p_l^*}^{\alpha_l}([a_l, b_l]^{p_l^*}, B)$ and define the underlying function space

$$\begin{aligned}\mathcal{H}(L, \boldsymbol{p}, \boldsymbol{p}^*, \alpha, B) = \{h = &h_L \circ h_{L-1} \circ \cdots \circ h_1 \circ h_0 : h_l = (h_{l1}, \cdots, h_{lp_{l+1}})^T, \\ &h_{lj} \in \mathcal{H}_{p_l^*}^{\alpha_l}([a_l, b_l]^{p_l^*}, B), \text{ for some } |a_l| \vee |b_l| \leq B\},\end{aligned} \tag{10}$$

where $\boldsymbol{p} = (p_0, \cdots, p_{L+1})$, $\boldsymbol{p}^* = (p_0^*, \cdots, p_L^*)$ and $\alpha = (\alpha_0, \cdots, \alpha_L)$. For this function space, we denote $\alpha_l^* = \alpha_l \prod_{i=l+1}^L (\alpha_i \wedge 1)$, $l^* = \arg\min_{l=1,\ldots,L}\{\alpha_l^*/(2\alpha_l^* + p_l^*)\}$, and

$$r_n = n^{-\frac{\alpha_{l^*}^*}{2\alpha_{l^*}^* + p_{l^*}^*}}. \tag{11}$$

Here $a \wedge b = \min(a, b)$. Without loss of generality, we assume that each feature has been rescaled to the unit interval, $X \in [0, 1]^p$. Our theoretical results stated below also depend on the following formal assumptions (proofs are relegated to the Appendix):

(A1) The underlying functions $h_1$ and $h_2$ in (4) belong to the Hölder class

$$\mathcal{H}_0 = \{h \in \mathcal{H}(L, \boldsymbol{p}, \boldsymbol{p}^*, \alpha, B) : \mathbb{E}[\Delta h(X)] = 0\}.$$

(A2) The observation times $O_i$ lie in a finite interval $[0, \tau]$, and the underlying baseline hazard $\lambda_0$ in (4) is positive with twice continuous derivative in $[0, \tau]$.

(A3) The sub-density $p(t) := d\mathbb{P}(O \leq t, \Delta = 1)/dt$ is twice differentiable in $[0, \tau]$ and there exists a positive constant $c$ such that $\mathbb{P}(T \geq \tau | X) \geq c$ with probability one.

(A4) The kernel function $K(\cdot)$, defined on $[-1, 1]$, is symmetric and twice-continuously differentiable in $(-1, 1)$.

(A5) $M = O(\log n)$, $s = O(nr_n^2 \log n)$, $d_1 \wedge \ldots \wedge d_M = O(nr_n^2)$ and $d_1 \vee \ldots \vee d_M = O(n)$.

Assumption (A1) is required for the identifiability of $h_1$ and $h_2$. Assumptions (A2) and (A3) are common smoothness assumptions for right censored data [58], where the study ends at time $\tau$. The second part of (A3) is typically satisfied in practice when the study must end while some subjects are still alive. Assumption (A4) ensures that the likelihood function can be well approximated by the pseudo-likelihood function. Assumption (A5) configures the structure of neural networks in (8) to ensure larger networks are used as sample size increases.

**Theorem 1** *Suppose assumptions (A1)-(A5) hold and $\rho s \to 0$ as $n \to \infty$ and $b = O(n^{-1/5})$. Then for any $\epsilon > 0$ there exists sufficiently large $A > 0$ and an estimator $\hat{\boldsymbol{h}} = (\hat{h}_1, \hat{h}_2) \in \mathcal{G}(\boldsymbol{d}, s, M, A) \times \mathcal{G}(\boldsymbol{d}, s, M, A)$ that optimizes (6) such that*

$$\lim_{n\to\infty} \mathbb{P}(d(\hat{\boldsymbol{h}}, \boldsymbol{h}) \geq \epsilon) = 0,$$

*where $\boldsymbol{h} = (h_1, h_2)$ and $d(\hat{\boldsymbol{h}}, \boldsymbol{h}) = \|\hat{h}_1 - h_1\|_{L^2([0,1]^p)} + \|\hat{h}_2 - h_2\|_{L^2([0,1]^p)}$. Moreover, we have*

$$\lim_{n \to \infty} \mathbb{P}(\|\hat{S}(\cdot|X) - S(\cdot|X)\|_{L^2([0,\tau])} \geq \epsilon) = 0.$$

The estimator $\hat{\boldsymbol{h}}_n = \hat{\boldsymbol{h}}(= (\hat{h}_1, \hat{h}_2))$ is based on the $n$ observations $\{X_i, O_i, \Delta_i\}_{i=1}^n$ in Section 2 and we omit its subscript $n$ for simplicity throughout the paper. From the proof of Theorem 1, we observe that both the approximation error and estimation error contributes to the error $\hat{\boldsymbol{h}} - \boldsymbol{h}$. The approximation error is the smallest distance between $\boldsymbol{h}$ and the neural networks space $\mathcal{G}(\boldsymbol{d}, s, M, A) \times \mathcal{G}(\boldsymbol{d}, s, M, A)$, for which more complex $\mathcal{G}(\boldsymbol{d}, s, M, A)$ yields smaller approximation errors. On the other hand, the estimation error is the difference between the minimizer $\hat{\boldsymbol{h}}$ and the optimal approximation of $\boldsymbol{h}$ in $\mathcal{G}(\boldsymbol{d}, s, M, A)$. With increasing sample sizes $n$, both the approximation error and estimation error become smaller. The next theorem shows the exact convergence rates of the proposed estimators.

**Theorem 2** *Let $\alpha_* = \lfloor \alpha_{l*}^* \rfloor + 1$. Under assumptions (A1)-(A5), if the baseline harzard function $\lambda_0$ is $\alpha_*$-th continuously differentiable, $b = O(n^{-1/(2\alpha_*+1)})$ and $\rho s = O(r_n^2)$, then we have*

$$d(\hat{\boldsymbol{h}}, \boldsymbol{h}) = O_p(r_n \log^2 n) \quad \text{and} \quad \|\hat{S}(\cdot|X) - S(\cdot|X)\|_{L^2([0,\tau])} = O_p(r_n \log^2 n).$$

Theorem 2 shows that the resulting convergence rates are jointly determined by the smoothness and intrinsic dimension of the true functions, which are similar to Schmidt-Hieber [53] for nonparametric regression and Zhong et al. [66] for the partially linear Cox model. For example, if the functions $h_1$ and $h_2$ have the form in (9) with twice differentiable functions $h_{kl}$, the convergence rate of the proposed method is of the order $n^{-2/7} \log n$, but classical nonparametric smoothing methods, such as kernel smoothing and B-splines, would yield a slower convergence rate of order $n^{-2/13}$. This illustrates how deep learning is able to alleviate the curse of dimensionality and identify the low dimensional structure of the data embedded in higher dimensions.

## 4 Experiments

This section presents the numerical performance of DeepEH in comparison with ten other survival methods on four real datasets. The Appendix contains additional empirical studies and details regarding our implementation.

### 4.1 Evaluation Criteria

**Concordance Index.** Due to the presence of censoring in survival data, traditional performance measures such as mean squared error cannot be used to evaluate the accuracy of predictions. Instead, the concordance-index (C-index, [27]) is one of the most widely used performance measures for survival models. It assesses how good a model is by measuring the concordance between the rankings of the predicted event times and the true event times. Specifically, if the predicted event time of the $i$-th individual is $\hat{T}_i$, the C-index is defined by $C = \mathbb{P}(\hat{T}_i < \hat{T}_j | O_i < O_j, \Delta_i = 1)$. However, it is difficult to obtain the predicted event time in most survival models, so the following C-index proposed in Antolini et al. [3] is often used in practice:

$$C = \mathbb{P}(\hat{S}(O_i|X_i) < \hat{S}(O_i|X_j)|O_i < O_j, \Delta_i = 1). \tag{12}$$

If $\{X_i, O_i, \Delta_i\}_{i=1}^n$ and $\hat{S}(t|X_i)$ denote observations and predicted conditional probabilities, respectively, the C-index in (12) can be estimated empirically by

$$\hat{C} = \frac{\sum_{i=1}^n \sum_{j=1}^n \Delta_i I(\hat{S}(O_i|X_i) < \hat{S}(O_i|X_j))}{\sum_{i=1}^n \sum_{j=1}^n \Delta_i I(O_i < O_j)}.$$

The range of the C-index is $[0, 1]$, and larger values indicate better performance with a random guess leading to a C-index of 0.5.

**Integrated Brier Score.** Similar to the mean squared error, the Brier score [7] is a measure of performance based on the predicted probability for a binary outcome. Graf et al. [24] extended the

Brier score to right censored data in order to assess the accuracy of an estimated survival function at some time $t$. Under right censorship, the Brier score at time $t$ is formally given by

$$BS(t) = \frac{1}{n} \sum_{i=1}^{n} \left\{ \frac{[\hat{S}(t|X_i)]^2 1(O_i \leq t)\Delta_i}{\hat{G}(O_i)} + \frac{[1 - \hat{S}(t|X_i)]^2 1(O_i > t)}{\hat{G}(t)} \right\},$$

where $\hat{G}$ is the Kaplan-Meier estimator of $G(t) := P(C > t)$ for censoring time $C$. As a metric to evaluate the performance of survival estimates, we consider the integrated Brier score (IBS)

$$IBS = \int_0^\tau BS(t) \, dt. \tag{13}$$

IBS is a proper scoring rule [23], where smaller values of this metric indicate better performance. We calculate IBS in our experiments via a Riemann sum of the Brier score $BS(t)$.

## 4.2 Baseline Methods

We compare our proposed models with three traditional survival methods, random survival forest and six recent deep learning methods.

**CoxPH**, **AFT**, **EH** represent the standard Cox proportional hazard, accelerated failure time and extended hazard models introduced in Section 2.

**RSF** leverages the accuracy of random forests for right-censored survival settings [33]. RSF bootstrap samples from the original data and grows a tree for each bootstrapped data set by splitting on nodes that maximize the log-rank difference between their sub-nodes. It then applies the Nelson–Aalen estimator to calculate the cumulative hazard function for each tree and finally averages these estimates.

**DeepSurv** replaces the linear relationship in the CoxPH model with a neural network [38].

**CoxTime** extends CoxPH with a time-dependent hazard ratio, learned using a neural network [41].

**PCHazard** treats the hazard function as piece-wise constant in predefined time-intervals, using a neural network to maximize this log-likelihood [40].

**DeepHit** directly estimates the joint distribution of events in discrete time, using a multi-task neural network trained with a likelihood loss and a ranking loss [43].

**DSM** develops a deep learning framework to estimate the survival function that can also handle competing risks in addition to single event survival data [47].

**DeepAFT** is adapted from the proposed DeepEH when enforcing $h_1 = h_2$, and we implement it as a special case of DeepEH.

Among all the methods considered in this paper, DeepHit is the most general model, but it may only perform well in applications where data are plentiful (which is not the case in existing survival studies) and no lower dimensional structures exist for the survival times.

## 4.3 Data Sets

Here we evaluate the performance of various survival methods using four real survival datasets: Chemotherapy for Stage B/C colon cancer (COLON), Molecular Taxonomy of Breast Cancer International Consortium (METABRIC), Rotterdam tumor bank and German Breast Cancer Study Group (RotGBSG) and Worcester Heart Attack Study (WHAS) and .

**COLON** was collected from a sample of 929 subjects to evaluate the effects of the drugs levamisole and fluorouracil on resected colon carcinoma [45]. By the end of study, 48.6 percent of patients were dead. Twelve features are included, among which two are continuous features: age, number of lymph nodes with detectable cancer, and ten are categorical features: treatment, sex, obstruction of colon by tumour, perforation of colon, adherence to nearby organs, differentiation of tumour, extent of local spread, time from surgery to registration, more than 4 positive lymph nodes.

**METABRIC** includes paired DNA-RNA profiles and clinical information of 1,980 breast cancer patients, among which 57.7 percent are recorded to die from the cancer [16]. Based on Immunohistochemical 4 score plus Clinical Treatment Score (IHC4+CTS, [42]), DeepSurv [38] considered four

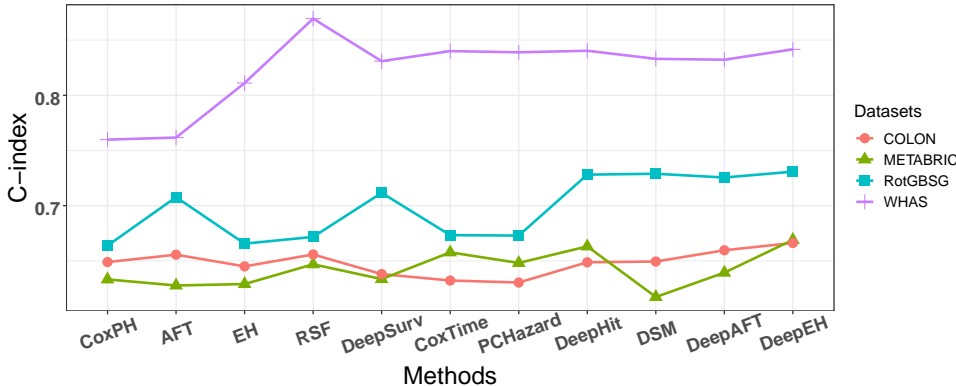

Figure 1: Average Concordance index achieved by each survival model (higher is better) over five different train/test splits of each dataset.

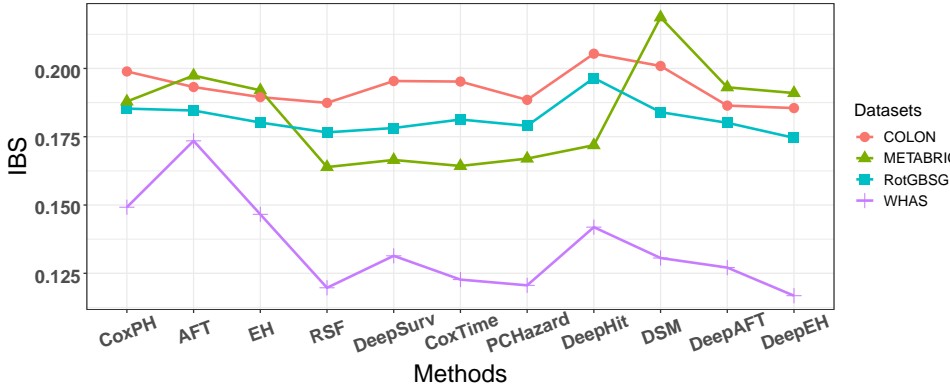

Figure 2: Average Integrated Brier Score achieved by each survival model (lower is better) over five different train/test splits of each dataset.

genes, four clinical indicators, and each patient's age as features to predict survival. We follow the same choices in our study. The four genes (MK167, EGFG, PGR and ERBB2) and the patient's age are continuous features, while hormone treatment indicator, radiotherapy indicator, chemotherapy indicator, ER-positive indicator are binary features.

**RotGBSG** is the breast cancer data extracted from the Rotterdam tumor bank [21] and the German Breast Cancer Study Group [54]. The data set consists of 2,232 patients and 56.7 percent of patients have observed death time. During the studies, researchers recorded the standard features: age, tumor size, number of positive lymph nodes, progesterone and oestrogen receptor status, menopausal status and tumor grade for each patient. In our analysis, tumor size, number of positive lymph nodes and tumor grade are coded to be ordinal features, whereas other features are coded to be continuous.

**WHAS** is a data set of 1,638 subjects for studying the factors of acute myocardial infarction [30]. The features contain sex, age, body mass index, left heart failure complications, and order of myocardial infarction, of which age and body mass index are continuous features and others are binary. By the end of the WHAS study, 42.1 percent of the subjects had died of acute myocardial infarction.

## 4.4 Results

The evaluation for each dataset is done via 5-fold cross-validation, where we randomly split each dataset into 5 equal folds that correspond to different train/test sets. Within each fold, 20% of the training data is reserved as a validation set. Performance is gauged via the C-index and IBS.

Figure 1 and Figure 2 show the average C-index and IBS (on test data) over five different train/test splits, respectively. More details, including the standard deviations, are displayed in Table 1 of the

Appendix. In terms of the C-index, DeepEH outperforms the other ten survival methods on three datasets: COLON, METABRIC and RotGBSG. For the WHAS dataset, RSF is the best method followed by DeepEH. We also observe that DeepEH improves over its classical predecessor, the EH model. In addition, DeepEH outperforms all its submodels, i.e., CoxPH, AFT, DeepSurv and DeepAFT, on all datasets. In terms of IBS, DeepEH performs the best among all eleven survival methods on COLON, RotGBSG and WHAS, and it outperformed the other ten methods substantially on RotGBSG. Similiar to C-index, DeepEH also has better performance in IBS than its submodels (CoxPH, AFT, EH, DeepSurv and DeepAFT) on all datasets.

## 5 Conclusion

Due to its massive potential, deep learning for survival analysis has attracted considerable attention across epidemiology and clinical research. A theoretical understanding, however, is still at its infancy. Mathematically and empirically, this work studied deep extensions of a flexible survival analysis method, the EH model, which can represent a wider family of survival models. We introduced a special but widely used neural network structure to facilitate mathematical study of deep survival analysis. Under the EH model, the neural network estimators of our proposed DeepEH method are asymptotically consistent, as are the corresponding estimators of the survival function. The favorable convergence rates of the new estimators reveal that deep learning is a powerful tool to mitigate the curse of dimensionality with fixed but high dimensional covariates. Empirically, the proposed DeepEH model achieves superior C-index and IBS than competing methods. Our methods thus constitute valuable additions to the practical survival modeling toolkit. Their broader impact will likely be to improve estimation accuracy in existing survival analysis applications.

In the future, practical modifications of the neural network architecture may be investigated to adapt the deep survival methods presented here for more complex features such as images [67] or gene expression measurements [10]. While all experiments in this paper were limited to time-invariant features, extensions to time-varying features (longitudinal data) remain of interest. Longitudinal covariates bring rich information to the study but pose challenges to conventional survival models, such as the CoxPH or AFT model, which require continuous measurements of the entire trajectory of the longitudinal covariates that are unavailable in practice [31, 61]. The (bio)statistics community has been actively researching how to best model survival data and longitudinal covariate processes jointly [18, 52, 64]. In particular, joint modeling for EH models with longitudinal covariates has been explored by Tseng et al. [58, 60]. Thus a natural direction for future work is to extend our DeepEH models to this setting. In addition, this paper only provides evaluations based on C-index and IBS, where the former metric measures the discriminative ability of a model while the latter accounts for both discrimination and calibration. Broader evaluation of deep survival methods remains of interest, including measures such as 1-calibration and D-calibration [25].

## Acknowledgements

We are grateful to anonymous Referees for valuable feedback on this work. The research of Qixian Zhong is supported by National Natural Science Foundation of China grant NSFC-11931001, Key Laboratory of Econometrics (Xiamen University), and Ministry of Education and Fujian Key Lab of Statistical Science, Xiamen University. The research of Jane-Ling Wang is supported by the USA NSF grant DMS-1914917.

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
