# Appendix: Deep Extended Hazard Models for Survival Analysis

**Qixian Zhong**
Department of Statistics and Data Science
School of Economics and Wang Yanan Institute for Studies in Economics (WISE)
Xiamen University
qxzhong@xmu.edu.cn

**Jonas Mueller**
Amazon Web Services
jonasmue@amazon.com

**Jane-Ling Wang**
Department of Statistics
UC Davis
janelwang@ucdavis.edu

## A  Proofs of Theorems

### A.1  Proof of Theorem 1

We consider using Theorem 5.7 of Van der Vaart [2000] to show the results of Theorem 1. For brevity, we denote

$$S_{0n}(u, g_1) = \frac{1}{n} \sum_{i=1}^{n} \Delta_i K_b(R_i(g_1) - u), \quad S_0(u, g_1) = \frac{d\mathbb{P}(\Delta = 1, R(g_1) \leq u)}{du},$$

and

$$S_{1n}(u, g_1, g_2) = \frac{1}{n} \sum_{i=1}^{n} \frac{e^{g_2(X_i)}}{e^{g_1(X_i)}} \int_{-\infty}^{R_i(g_1)-u} K_b((v)) dv,$$

$$S_1(u, g_1, g_2) = \mathbb{E}[e^{g_2(X) - g_1(X)} I(R(g_1) \geq u)].$$

Then the pseudo-likelihood function is expressed as

$$\mathcal{L}_n(g_1, g_2) = \frac{1}{n} \sum_{i=1}^{n} \Delta_i g_2(X_i) - \frac{1}{n} \sum_{i=1}^{n} \Delta_i R_i(g_1) + \frac{1}{n} \sum_{i=1}^{n} \Delta_i \log \left\{ \frac{S_{0n}(u, g_1)}{S_{1n}(u, g_1, g_2)} \Big|_{u=R_i(g_1)} \right\}.$$

Let $\mathcal{G} = \{g \in \mathcal{G}(\boldsymbol{d}, s, M, A) : \mathbb{E}[\Delta g(X)] = 0\}$ and

$$\mathcal{L}(g_1, g_2) = \mathbb{E}\left[ \Delta g_2(X) - \Delta R(g_1) + \Delta \log \left\{ \frac{S_0(u, g_1)}{S_1(u, g_1, g_2)} \Big|_{u=R(g_1)} \right\} \right].$$

By Lemma 2.4 of Schuster [1969], Lemma 2.4.3 of Van der Vaart and Wellner [1996] and Lemma 5 of Schmidt-Hieber [2020], we have

$$\sup_{g_1, g_2 \in \mathcal{G}, u \in \mathbb{R}} |S_{0n}(u, g_1) - S_0(u, g_1)| \to 0, \text{a.s.,}$$

35th Conference on Neural Information Processing Systems (NeurIPS 2021).

and
$$\sup_{g_1,g_2\in\mathcal{G},u\in\mathbb{R}}|S_{1n}(u,g_1,g_2)-S_1(u,g_1,g_2)|\to 0, \text{ a.s.},$$

which further implies that
$$\sup_{g_1,g_2\in\mathcal{G}}|\mathcal{L}_n(g_1,g_2)-\mathcal{L}(g_1,g_2)|\to 0,\text{ as }n\to\infty. \tag{1}$$

Next, we show that, with the true functions $h_1$ and $h_2$, for any $\delta>0$,
$$\sup_{\substack{g_1,g_2\in\mathcal{G},\\ \|g_1-h_1\|_{L^2}+\|g_2-h_2\|_{L^2}\geq\delta}}\mathcal{L}(g_1,g_2)<\mathcal{L}(h_1,h_2). \tag{2}$$

For any $g_1,g_2\in\mathcal{G}$, define $\Psi(u,v;g_1,g_2)=\mathcal{L}(h_1+u(g_1-h_1),h_2+v(g_2-h_1))$. Then by Taylor expansion and assumptions (A1) and (A2), we get
$$\Psi(u,v;g_1,g_2)-\Psi(0,0;g_1,g_2)\asymp -\mathbb{E}\{[u(g_1-h_1)]^2+[v(g_1-h_1)]^2\}, \tag{3}$$

where $a_n\asymp b_n$ means that there exists $c_1,c_2>0$ such that $c_1a_n\leq b_n\leq c_2a_n$. Hence it implies (2) if we let $u=v=1$.

Furthermore, Let
$$\bar{h}_1=\arg\min_{g_1\in\mathcal{G}}\|g_1-h_1\|_{L^2}\text{ and }\bar{h}_2=\arg\min_{g_2\in\mathcal{G}}\|g_2-h_2\|_{L^2}.$$

Then, by (1), (3) and Theorem 5 in Schmidt-Hieber [2020], it follows that
$$\begin{aligned}|\mathcal{L}_n(\bar{h}_1,\bar{h}_2)-\mathcal{L}_n(h_1,h_2)|\leq & |\mathcal{L}_n(\bar{h}_1,\bar{h}_2)-\mathcal{L}(\bar{h}_1,\bar{h}_2)|+|\mathcal{L}(h_1,h_2)-\mathcal{L}_n(h_1,h_2)|\\ & +|\mathcal{L}(\bar{h}_1,\bar{h}_2)-\mathcal{L}(h_1,h_2)|\\ \lesssim & o_p(1)+o_p(1)+\mathbb{E}[(\bar{h}_1-h_1)^2+(\bar{h}_2-h_2)^2]\\ = & o_p(1),\text{ as }n\to\infty.\end{aligned} \tag{4}$$

where $a_n\lesssim b_n$ means that there exists $c_1>0$ such that $a_n\leq c_1b_n$.

The the definition of $(\hat{h}_1,\hat{h}_2)$ and (4) gives
$$\begin{aligned}\mathcal{L}_n(\hat{h}_1,\hat{h}_2)\geq & \mathcal{L}_n(\bar{h}_1,\bar{h}_2)-\rho[J(\bar{h}_1,\bar{h}_2)-J(\hat{h}_1,\hat{h}_2)]\\ = & \mathcal{L}_n(h_1,h_2)-o_p(1),\end{aligned} \tag{5}$$

where $|\rho[J(\bar{h}_1,\bar{h}_2)-J(\hat{h}_1,\hat{h}_2)]|\leq 2\rho s\to 0$, as $n\to\infty$.

Therefore, combining (1), (2) and (5), Theorem 5.7 of Van der Vaart [2000] implies that, for any $\epsilon>0$,
$$\mathbb{P}(\|(\hat{h}_1,\hat{h}_2)-(h_1,h_2)\|_{L^2}\geq\epsilon)\to 0\text{ as }n\to\infty.$$

For the consistence of the survival estimator $\hat{S}(\cdot|X)$. Denote $\kappa=\max_{x\in[0,1]^p}\tau e^{h_1(x)}$. Let
$$R_{0n}(v,g_1)=\frac{1}{nv}\sum_{i=1}^n\Delta_i K_b(R_i(g_1)-\log v),\quad R_0(v,g_1)=\frac{d\mathbb{P}(\Delta=1,Oe^{g_1(X)}\leq v)}{dv}$$

and
$$R_{1n}(v,g_1,g_2)=\frac{1}{n}\sum_{i=1}^n\frac{e^{g_2(X)}}{e^{g_1(X)}}\int_{-\infty}^{R_i(g_1)-\log v}K_b(v)dv,$$
$$R_1(v,g_1,g_2)=\mathbb{E}[e^{g_2(X)-g_1(X)}I(Oe^{g_1(X)}\geq v)].$$

Then
$$\hat{\lambda}_0(v)-\lambda_0(v)=\frac{R_{0n}(v,\hat{h}_1)}{R_{1n}(v,\hat{h}_1,\hat{h}_2)}-\frac{R_0(v,h_1)}{R_1(v,h_1,h_2)}.$$

Note that
$$\frac{R_{0n}(v,\hat{h}_1)}{R_{1n}(v,\hat{h}_1,\hat{h}_2)}-\frac{R_0(v,h_1)}{R_1(v,h_1,h_2)}=\frac{R_{0n}(v,\hat{h}_1)R_1(v,h_1,h_2)-R_{1n}(v,\hat{h}_1,\hat{h}_2)R_0(v,h_1)}{R_{1n}(v,\hat{h}_1,\hat{h}_2)R_1(v,h_1,h_2)}.$$

The denominator $R_{1n}(v, \hat{h}_1, \hat{h}_2)R_1(v, h_1, h_2)$ is asymptotically bounded away from zero by Assumption (A3), and the numerator is

$$R_{0n}(v, \hat{h}_1)R_1(v, h_1, h_2) - R_{1n}(v, \hat{h}_1, \hat{h}_2)R_0(v, h_1)$$
$$= [R_{0n}(v, \hat{h}_1) - R_0(v, h_1)]R_1(v, h_1, h_2) - [R_{1n}(v, \hat{h}_1, \hat{h}_2) - R_1(v, h_1, h_2)]R_0(v, h_1).$$

By assumptions (A2) and (A3), we further observe that

$$\|R_{0n}(\cdot, \hat{h}_1) - R_0(\cdot, h_1)\|_{L^2([0,\kappa])}$$
$$\leq \|R_{0n}(\cdot, \hat{h}_1) - R_0(\cdot, \hat{h}_1)\|_{L^2([0,\kappa])} + \|R_0(\cdot, \hat{h}_1) - R_0(\cdot, h_1)\|_{L^2([0,\kappa])}$$
$$\to 0, \text{ as } n \to \infty.$$

Following similar discussions, we also have

$$\|R_{1n}(\cdot, \hat{h}_1, \hat{h}_2) - R_1(\cdot, h_1, h_2)\|_{L^2([0,\kappa])} \to 0, \text{ as } n \to \infty.$$

And $R_0(v, h_1)$ and $R_1(v, h_1, h_2)$ are asymptotically bounded. Thus, by Lebesgue's dominated convergence theorem, we have

$$\left\| \frac{R_{0n}(\cdot, \hat{h}_1)}{R_{1n}(\cdot, \hat{h}_1, \hat{h}_2)} - \frac{R_0(\cdot, h_1)}{R_1(\cdot, h_1, h_2)} \right\|_{L^2([0,\kappa])} \to 0, \text{ as } n \to \infty,$$

which implies that, for any $\epsilon > 0$, $\lim_{n\to\infty} \mathbb{P}(\|\hat{\lambda}_0 - \lambda_0\|_{L^2([0,\kappa])} \geq \epsilon) = 0$.

Furthermore, by

$$\hat{\Lambda}(u) - \Lambda_0(u) = \int_0^u \{\hat{\lambda}_0(v) - \lambda_0(v)\}dv,$$

we have

$$\|\hat{\Lambda} - \Lambda_0\|_{L^2([0,\kappa])} \leq \kappa \|\hat{\lambda}_0 - \lambda_0\|_{L^2([0,\kappa])} \to 0, \text{ as } n \to \infty.$$

This gives $\lim_{n\to\infty} \mathbb{P}(\|\hat{\Lambda}_0 - \Lambda_0\|_{L^2([0,\kappa])} \geq \epsilon) = 0$, for any $\epsilon > 0$. And by combining Theorem 1, we obtain, for any $\epsilon > 0$,

$$\lim_{n\to\infty} \mathbb{P}(\|\hat{S}(\cdot|X) - S(\cdot|X)\|_{L^2([0,\tau])} \geq \epsilon) = 0.$$

## A.2 Proof of Theorem 2.

Denote $\mathcal{A}_\delta = \{\mathbf{g} = (g_1, g_2) \in \mathcal{G} \times \mathcal{G} : \delta/2 \leq d(\mathbf{g}, \mathbf{h}) \leq \delta\}$ and

$$\tilde{\mathcal{L}}_n(g_1, g_2) = \frac{1}{n}\sum_{i=1}^n \Delta_i g_2(X_i) - \frac{1}{n}\sum_{i=1}^n \Delta_i R_i(g_1) + \frac{1}{n}\sum_{i=1}^n \Delta_i \log\left\{ \frac{S_0(u, g_1)}{S_1(u, g_1, g_2)}\Big|_{u=R_i(g_1)} \right\}.$$

Note that, when $b = O(n^{-1/(2\alpha_*+1)})$,

$$\mathbb{E}\{\sup_{\mathbf{g}\in\mathcal{A}_\delta} |[\mathcal{L}_n(\mathbf{g}) - \tilde{\mathcal{L}}_n(\mathbf{g})] - [\mathcal{L}_n(\mathbf{h}) - \tilde{\mathcal{L}}_n(\mathbf{h})]|\} \lesssim \delta n^{-\frac{\alpha_*}{2\alpha_*+1}}[\log(n)]^{1/2}. \tag{6}$$

Then, we show that

$$\mathbb{E}\{\sup_{\mathbf{g}\in\mathcal{A}_\delta} \sqrt{n}|[\tilde{\mathcal{L}}_n(\mathbf{g}) - \mathcal{L}(\mathbf{g})] - [\tilde{\mathcal{L}}_n(\mathbf{h}) - \mathcal{L}(\mathbf{h})]|\} \lesssim \varphi_n(\delta), \tag{7}$$

where $\varphi_n(\delta) = \delta\sqrt{s\log\frac{U}{\delta}} + \frac{s}{\sqrt{n}}\log\frac{U}{\delta}$ with $U = K\prod_{k=0}^K(d_k+1)\sum_{k=0}^K d_k d_{k+1}$. Denote $\mathbb{G}_n = \sqrt{n}(\mathbb{E}_n - \mathbb{E})$, where $\mathbb{E}_n X = 1/n\sum_{i=1}^n X_i$. Let $\psi_1(\mathbf{g}) = \Delta[-R(g_1) + g_2]$ and $\psi_2(\mathbf{g}) = \Delta\log\{[S_0(u, g_1)/S_1(u, g_1, g_2)]|_{u=R_i(g_1)}\}$, then

$$\sqrt{n}\{[\tilde{\mathcal{L}}_n(\mathbf{g}) - \mathcal{L}(\mathbf{g})] - [\tilde{\mathcal{L}}_n(\mathbf{h}) - \mathcal{L}(\mathbf{h})]\} = \mathbb{G}_n[\psi_1(\mathbf{g}) - \psi_1(\mathbf{h})] + \mathbb{G}_n[\psi_2(\mathbf{g}) - \psi_2(\mathbf{h})].$$

Let $\mathcal{F}_\delta = \{\psi_1(\mathbf{g}) - \psi_1(\mathbf{h}) : \mathbf{g} \in \mathcal{A}_\delta\}$. For any $\mathbf{g}_1, \mathbf{g}_2 \in \mathcal{A}_\delta$, we have

$$\mathbb{E}[\psi_1(\mathbf{g}_1) - \psi_1(\mathbf{g}_2)]^2 \lesssim d^2(\mathbf{g}_1, \mathbf{g}_2).$$

By Lemma 5 of Schmidt-Hieber [2020], it follows that

$$\mathcal{N}_{[\,]}(\epsilon, \mathcal{F}_\delta, L^2(P)) \lesssim s \log\left(\frac{U}{\epsilon}\right),$$

where $\mathcal{N}_{[\,]}$ is the bracketing number. This implies the bracketing integral

$$J_{[\,]}(\delta, \mathcal{F}_\delta) := \int_0^\delta \sqrt{1 + \mathcal{N}_{[\,]}(\epsilon, \mathcal{F}_\delta, L^2(P))} d\epsilon \lesssim \delta\sqrt{s \log\left(\frac{U}{\delta}\right)}.$$

Then we obtain

$$\mathbb{E}\Big\{ \sup_{\mathbf{g} \in \mathcal{A}_\delta} |\mathbb{G}_n[\psi_1(\mathbf{g}) - \psi_1(\boldsymbol{h})]| \Big\} \lesssim J_{[\,]}(\delta, \mathcal{F}_\delta)\Big[1 + \frac{J_{[\,]}(\delta, \mathcal{F}_\delta)}{\delta^2 \sqrt{n}}\Big] \lesssim \delta\sqrt{s \log\left(\frac{U}{\delta}\right)} + \frac{s}{\sqrt{n}}\log\left(\frac{U}{\delta}\right).$$

Likewise, by Taylor expansion, it follows that

$$\mathbb{E}\Big\{ \sup_{\mathbf{g} \in \mathcal{A}_\delta} |\mathbb{G}_n[\psi_2(\mathbf{g}) - \psi_2(\boldsymbol{h})]| \Big\} \lesssim \delta\sqrt{s \log\left(\frac{U}{\delta}\right)} + \frac{s}{\sqrt{n}}\log\left(\frac{U}{\delta}\right).$$

Hence (7) holds.

On the other hand, an easy calculation yields

$$\varphi(r_n \log^2 n) \lesssim \sqrt{n}(r_n \log^2 n)^2. \tag{8}$$

By (3), (6), (7) and the definition of $\hat{\boldsymbol{h}}$, we get

$$\mathcal{L}_n(\hat{\boldsymbol{h}}) \geq \mathcal{L}_n(\boldsymbol{h}) - O_p((r_n \log^2 n)^2). \tag{9}$$

Therefore, $d(\hat{\boldsymbol{h}}, \boldsymbol{h}) = O_p(r_n \log^2 n)$ by (6), (7), (8) and (9) with an application of Theorem 3.4.1 in Van der Vaart and Wellner [1996].

Furthermore, similar to the proof in Theorem 1, we get

$$\|\hat{S}(\cdot|X) - S(\cdot|X)\|_{L^2([0,\tau])} = O_p(r_n \log^2 n).$$

## B    Additional Results and Simulation Studies

As a finer-grained presentation of Figures 1-2, Table 1 shows the C-index and IBS achieved on each real dataset by various survival methods (both mean and standard deviation).

To further understand the empirical performance of our methods in a controlled setting, we also apply them to synthetically generated datasets. For each simulation study, we generate datasets with sample size $n = 1,000, 2,000, 4,000$ and use these datasets to compare EH and DeepEH. We first define:

$$h_1(x) = \frac{1}{5}\big(x_1 - x_2 + x_3 - x_4 + x_5 + x_6 - 5\big)$$

$$h_2(x) = \frac{1}{5}\big(x_1 + x_2 + x_3 + x_4 + x_5 + x_6 - 8\big)$$

$$h_3(x) = \frac{1}{5}\big(x_1^2 - x_2^2 + x_3^2 - x_4^2 + x_5^2 + x_6 - 5\big)$$

$$h_4(x) = \frac{1}{5}\big(x_1^2 + x_2^2 + x_3^3 + x_4^4 + x_5^5 + x_6 - 8\big)$$

Here, we consider 6-dimensional covariates, with continuous features $X_1, \cdots, X_5$ uniformly distributed over the unit interval, and $X_6 \sim \text{Bernoulli}(p = 0.5)$ as a binary feature. Based on this feature distribution, we sample data from four different underlying models:

*Case 1*: The event times are distributed according to the following hazard function:

$$\lambda(t|X = x) = \lambda_0(te^{h_1(x)})e^{h_2(x)}$$

Table 1: C-index and IBS achieved on test data by various methods in four datasets. Results listed here are the mean and standard deviation (in parenthesis) of the C-index and IBS over five different train/test splits of each dataset (best performance in bold).

| Methods | C-index | | | | IBS | | | |
|---|---|---|---|---|---|---|---|---|
| | RotGBSG | METABRIC | WHAS | COLON | RotGBSG | METABRIC | WHAS | COLON |
| CoxPH | 0.6638 (0.0111) | 0.6332 (0.0131) | 0.7598 (0.0120) | 0.6489 (0.0411) | 0.1853 (0.0046) | 0.1879 (0.0126) | 0.1492 (0.0051) | 0.1989 (0.0115) |
| AFT | 0.7077 (0.0512) | 0.6277 (0.0189) | 0.7617 (0.0153) | 0.6556 (0.0247) | 0.1846 (0.0091) | 0.1974 (0.0148) | 0.1735 (0.0037) | 0.1932 (0.0071) |
| EH | 0.6656 (0.0418) | 0.6290 (0.0187) | 0.8111 (0.0299) | 0.6450 (0.0267) | 0.1802 (0.0159) | 0.1920 (0.0131) | 0.1466 (0.0048) | 0.1895 (0.0064) |
| RSF | 0.6717 (0.0130) | 0.6469 (0.0157) | **0.8694** (0.0179) | 0.6558 (0.0210) | 0.1766 (0.0037) | **0.1639** (0.0057) | 0.1197 (0.0028) | 0.1874 (0.0107) |
| DeepSurv | 0.7116 (0.0096) | 0.6335 (0.0147) | 0.8309 (0.1554) | 0.6380 (0.0431) | 0.1782 (0.0043) | 0.1665 (0.0070) | 0.1314 (0.0330) | 0.1954 (0.0121) |
| CoxTime | 0.6733 (0.0091) | 0.6577 (0.0122) | 0.8400 (0.0115) | 0.6322 (0.0483) | 0.1813 (0.0025) | 0.1643 (0.0078) | 0.1227 (0.0047) | 0.1952 (0.0149) |
| PCHazard | 0.6729 (0.0131) | 0.6481 (0.0167) | 0.8389 (0.0101) | 0.6304 (0.0380) | 0.1790 (0.0039) | 0.1670 (0.0069) | 0.1206 (0.0062) | 0.1885 (0.0380) |
| DeepHit | 0.7281 (0.0168) | 0.6631 (0.0159) | 0.8403 (0.0168) | 0.6487 (0.0426) | 0.1965 (0.0028) | 0.1719 (0.0068) | 0.1419 (0.0058) | 0.2054 (0.0063) |
| DSM | 0.7289 (0.0630) | 0.6173 (0.0167) | 0.8330 (0.0347) | 0.6494 (0.0436) | 0.1840 (0.0020) | 0.2187 (0.0047) | 0.1306 (0.0117) | 0.2009 (0.0143) |
| DeepAFT | 0.7255 (0.0277) | 0.6393 (0.0143) | 0.8322 (0.0117) | 0.6596 (0.0334) | 0.1801 (0.0049) | 0.1931 (0.0100) | 0.1271 (0.0061) | 0.1864 (0.0113) |
| DeepEH | **0.7308** (0.0266) | **0.6690** (0.0129) | 0.8416 (0.0108) | **0.6662** (0.0371) | **0.1746** (0.0040) | 0.1910 (0.0084) | **0.1168** (0.0032) | **0.1855** (0.0101) |

with baseline hazard function:

$$\lambda_0(t) = 0.05e^t \tag{10}$$

The data in this case thus stem from a classical EH model. The censoring time $C$ are randomly generated uniformly from the interval $[0, 15.6]$ for each individual, which results in an overall censoring rate of $= 0.45$.

*Case 2*: Here $\lambda_0$ is the same as in (10). The hazard function of the event time is

$$\lambda(t|X = x) = \lambda_0(te^{h_3(x)})e^{h_4(x)}$$

The censoring times $C$ are from a uniform distribution on [0,15], which leads to an overall censoring rate of $0.46$. The data in this case thus stem from a DeepEH model.

We perform $Q = 100$ simulation runs for each sample size $n$. Let $\mathcal{D}^{(q)} = \{X_i^{(q)}, O_i^{(q)}, \Delta_i^{(q)}\}_{i=1}^n$ be the $q$-th dataset among the replicated runs. Since the true survival functions are known, rather the integrated Brier score, we consider the relative integrated squared error (RISE):

$$\text{RISE}^{(q)} = \underset{k}{\text{ave}} \left\{ \frac{\|\hat{S}^{(q)}(\cdot|X_k) - S(\cdot|X_k)\|_{L^2([0,\tau_q])}}{\|S(\cdot|X_k)\|_{L^2([0,\tau_q])}} \right\},$$

where the average is taken over a held-out test set from five-fold cross-validation, and $\tau_q$ is the maximum observed time of the test set. We also report the concordance index (C-index) produced by all methods.

Table 2 reports the RISE and C-index achieved by EH and DeeepEH methods in each simulation case. The survival function estimates and C-index are clearly improved as the sample size $n$ increases from $1,000$ to $4,000$. In addition, the proposed DeepEH performs better than classical EH in *Case 2* while they are comparable when the underlying model is *Case 1*.

Table 2: The relative integrated squared error (RISE) of estimated survival function and C-index for simulation studies with EH and DeepEH methods. We report the average and standard deviation (in parentheses) over 100 simulated datasets from each underlying model. In each simulation run, $80\%$ of the data were used for estimation, and $20\%$ are held out to evaluate RISE.

|  | $n$ | RISE | | C-index | |
| --- | --- | --- | --- | --- | --- |
|  |  | EH | DeepEH | EH | DeepEH |
| *Case 1* | 1,000 | 0.2510 (0.0153) | 0.2533 (0.0160) | 0.5693 (0.0315) | 0.5661 (0.0328) |
|  | 2,000 | 0.2261 (0.0132) | 0.2271 (0.0143) | 0.5662 (0.0234) | 0.5655 (0.0235) |
|  | 4,000 | 0.2251 (0.0127) | 0.2259 (0.0129) | 0.5698 (0.0227) | 0.5691 (0.0229) |
| *Case 2* | 1,000 | 0.1964 (0.0280) | 0.1958 (0.0257) | 0.5585 (0.0337) | 0.5777 (0.0323) |
|  | 2,000 | 0.1827 (0.0218) | 0.1796 (0.0203) | 0.5655 (0.0260) | 0.5838 (0.0253) |
|  | 4,000 | 0.1752 (0.0159) | 0.1698 (0.0139) | 0.5668 (0.0162) | 0.5885 (0.0142) |

## C  Implementation Details

As there are relatively few missing values in each real survival dataset used in our experiments (the highest proportion of missing values among the four datasets is 4.4%), we simply omit subjects with at least one missing feature. We employ one hot encoding for categorical features, and standardize the continuous and ordinal features with zero mean and a standard deviation of one.

We use a Gaussian kernel as our kernel function $K(\cdot)$, so the function $\Phi(t) = \int_0^t K(s)ds$ is simply the cumulative distribution function of the standard normal distribution. Theoretically, the optimal order of the bandwidth is $O((8\sqrt{2}/3)^{1/5}n^{-1/5})$ for kernel smoothing methods [Sheather and Jones, 1991]. In our experiments, we pre-select the bandwidth to be $(8\sqrt{2}/3)^{1/5}n^{-1/5}$. The results of baseline methods are obtained by running the code provided in the corresponding papers, including the choice of hyperparameters.

Implemented in PyTorch, our neural networks are simple multilayer perceptrons with the same number of neurons in every hidden layer and the same activation function between layers. Training was performed using the AdamW optimizer [Loshchilov and Hutter, 2017] with $L_2$ regularization for up to 5000 epochs, with early stopping based on pseudo-likelihoods evaluated on the validation data. The activation function we employed is Rectified Linear Units. Based on the pseudo-likelihood of the validation data, we used *grid search* select hyperparameters such as learning rate, weight decay, number of neurons and hidden layers, dropout rate. The search space is as follows:

- Learning rate: $0.0005, 0.0008, 0.001, 0.005, 0.01$;
- Weight decay: $0.0005, 0.005, 0.05, 0.1$;
- Number of neuron: $32, 64, 128, 256, 512$;
- Number of hidden layer: $1, 2, 3, 5$;
- Dropout rate: $0, 0.2, 0.4, 0.6$.

The implementations of other deep survival methods are based on publicly available codes from the corresponding papers. All of our experiments were carried out on a standard MacBook Pro laptop (Processor: 3.5 GHz Intel Core i7, Memory: 16 GB 2133 MHz LPDDR3).