# OpenReview forum: "Deep Extended Hazard Models for Survival Analysis"
_NeurIPS.cc/2021/Conference — NeurIPS 2021 Poster_

### Official Review · Reviewer_z47A · 2021-07-02

**Rating:** 5
**Confidence:** 3

**Summary:**

Substituting two general functions for linear risk predictors in the Extended Hazard (EH) model, the DeepEH approximates these two functions via feed-forward neural networks (FNN). The DeepEH may fall into the category of FNN-assisted survival analysis which utilizes FNNs to estimate the hazard/survival function. The shining point of this paper is the established asymptotic properties of estimators from DeepEH.




**Limitations And Societal Impact:**

The extension from the EH model is natural. In addition, there has been literature that proves the power of FNN from a theoretical point of view, whereas this paper fails to make a review in this regard. Among other works, Schmidt-Hieber (2020) gave an exact upper bound of the approximation error for FNNs involving the least-square loss. Since the DeepEH optimizes a likelihood-based loss, this paper builds up its asymptotic properties by following assumptions and proofs of Theorems 1 and 2 in Schmidt-Hieber (2020) as well as theories on empirical processes.

Additional Feedback: 1) In the manuscript, $\mathbb{P}$ mostly represents a probability but sometimes for a cumulative distribution function (e.g., Eqs. (3) and (4) and L44, all in Appendix), which leads to confusion. 2). The notation $K$ is abused too: it is used both for a known kernel function (e.g., L166) and the number of layers (e.g., L176). 3). What is $K_b$ in estimating baseline hazard (L172)?

**Main Review:**

Depending on up-to-date theoretical developments on FNN approximation such as Schmidt-Hieber (2020, Ann. Statist. 48:1875--1897), this submission makes a theoretical contribution to the FNN-assisted survival analysis. We guess this effort is of interest to the community of deep learning research. Also, this paper involves extensive numerical comparison among DeepEH and a dozen competitors.

Correctness: We believe the theories and proofs are generally correct.

Clarity: Most parts are easy to follow.


**Time Spent Reviewing:**

1 hour

---

> ### Author Response · Authors · 2021-08-10
> **Response to Reviewer z47A**
>
> We thank you for your comments and provide our response below.
>
> > The extension from the EH model is natural. In addition, there has been literature that proves the power of FNN from a theoretical point of view, whereas this paper fails to make a review in this regard. Among other works, Schmidt-Hieber (2020) gave an exact upper bound of the approximation error for FNNs involving the least-square loss.
>
> Due to the page limit and our focus on survival modeling, we mainly review existing work in survival analysis rather than deep learning theory for supervised learning. We will change the statement "In this paper, we provide theoretical support for Deep Extended Hazard (DeepEH) model, ..." in line 86 to "Riding on the success of neural networks approximating to a nonparametric function [1,2,3,4,5], this paper provides a theoretical support for Deep Extended Hazard (DeepEH) model,..."
>
> References:
>
> $[1]$ Bauer, B. and Kohler, M. (2019). On deep learning as a remedy for the curse of dimensionality in nonparametric regression. Ann. Statist. 47 2261–2285.
>
> $[2]$ Cybenko, G. (1989). Approximation by superpositions of a sigmoidal function. Math. Control Signals Systems 2 303–314.
>
> $[3]$ Hornik, K., Stinchcombe, M., White, H. (1989). Multilayer feedforward networks are universal approximators. Neural Netw. 2 359–366.
>
> $[4]$ Yarotsky, D. (2017). Error bounds for approximations with deep ReLU networks. Neural Netw. 94 103–114.
>
> $[5]$ Schmidt-Hieber, J. (2020). Nonparametric regression using deep neural networks with ReLU activation function. Ann. Statist. 48 1875–1897.
>
> > Since the DeepEH optimizes a likelihood-based loss, this paper builds up its asymptotic properties by following assumptions and proofs of Theorems 1 and 2 in Schmidt-Hieber (2020) as well as theories on empirical processes.
>
> We emphasize that our extensions of NN approximation error theory to survival settings constitutes a nontrivial extension of the existing theory. Although we did utilize the result in Theorem 1 of  Schmidt-Hieber (2020), our results contain major differences. First, the target for the DeepEH model is the unobserved hazard function, not the observed scalar response considered in Schmidt-Hieber (2020). Second, our survival model contains three nonparametric components, $\lambda_0$, $h_1$ and $h_2$, while the regression model in Schmidt-Hieber (2020) contains only one nonparametric component. Third, the standard regression model considered by in Schmidt-Hieber only studies estimation of the conditional mean. In contrast, our model involves estimating a conditional distribution, which is much harder than the least squares problem of mean estimation. Fourth, we are dealing with censored data, which adds another layer of complexity that must be resolved in our theorems. Finally, besides the theoretical contribution in this paper, we also provide a practical new model that outperforms state-of-the-art neural survival models.
>
>
> > What is $K_b$ in estimating baseline hazard (L172)?
>
> We will clarify $K_b(x)= K(x/b)/b$ when it first appears in line 172.
>
>
>
> > In the manuscript, \mathbb{P} mostly represents a probability but sometimes for a cumulative distribution function (e.g., Eqs. (3) and (4) and L44, all in Appendix), which leads to confusion.
>
>
>  We followed the common notation in empirical processes that denotes \mathbb{P} both for the expectation and probability. Since this may be confusing to some, we will change the notation for expectations to  \mathbb{E} and keep  probability with \mathbb{P}.
>
>
>
> > The notation $K$ is abused too: it is used both for a known kernel function (e.g., L166) and the number of layers (e.g., L176).
>
>
>
> Indeed this is confusing so we will keep $K(\cdot)$ as the kernel function and change the number of layers $K$ to $M$, which has not been used in this paper.

---

### Official Review · Reviewer_hDJX · 2021-07-13

**Rating:** 6
**Confidence:** 4

**Summary:**

This paper introduces the Deep Extended Hazard Model as a new approach to learning a survival model, based on learning 2 neural nets, to estimate the 2 terms of the extended hazard function.  It explains how this model extends the EH model, which extends both the Cox proportional hazard model and Accelerated Failure Time.  It then provides first a theoretical analysis, following by an empirical exploration of these models

**Ethical Concerns:**

No ethical concerns.

**Limitations And Societal Impact:**

Predicting a patient's survival time for a given treatment, could help identify the appropriate treatment.  Of course, it could also be used to determine the most toxic poison or disease.  However, I do not think the manuscript needs to discuss this.



**Main Review:**

The paper did a nice job of motivating then explaining the various models considered. However, I had a hard time following the theoretical analysis.  Eg. I wish the authors had provided some details of the derivation of Eq1, or a pointer.  I also wondered which parts of Sec 3.2 were general for all neural nets, and what was specific to survival functions, and what just for variants of nn extensions of EH models? Some parts could have been a bit more precise--e.g I assume that \hat-h is based on the sample size n (line 225). Also, I did not understand how the theoretical analysis informed the development of the algorithm, or was used in the analysis of the empirical results.

They empirically explored many (10) survival models, over 4 different datasets. The authors should have performed some statistical tests, to indicate which of the results were significant.  Given the paper’s concern with high dimensional covariates, it was surprising that the datasets were all low-dimensional.  I wish the authors had empirically explored this issue, by using high dimensional data.

While IBS is considered both discriminative and calibration, it would have been useful to see if each dataset was calibrated--perhaps at the median time.  See the analysis in https://jmlr.csail.mit.edu/papers/v21/18-772.html .


wrt the IBS:  Li258, \hat-G is NOT the KM estimator.  It is the estimator of the conditional survival function of the censoring times calculated using the Kaplan-Meier method.  Is the integral Eq12 done analytically for your models, or estimated empirically? If the latter, are there issues related to the intervals considered?  Is \tau the largest time (censored or not), or the largest uncensored time, or something else?

The authors should have discussed how they split the data into 5 folds–was this balanced for (censored, death) times? For anything else?

I wish they plotted the results in a figure–perhaps moving Table 1 to the appendix. Similarly, it would be helpful if they had plotted the hazard curves, or the survival curves, to help the reader visualize the results.


li55: given the many different ways to evaluate a survival model, what does “improvement” mean?

li77: I assume  “omitted features” means some values are missing for some instances. If so, are these omissions considered  MCAR?

li119: extra β

li277: Why mention competing risks? Isn’t there only a single type of risk?

--------------
My assessment would be better if the paper

 included statistical conclusions

 explicitly connected the theoretical analysis to the algorithm design and/or the empirical results




**Time Spent Reviewing:**

6

---

> ### Author Response · Authors · 2021-08-10
> **Response to Reviewer hDJX**
>
> We thank you for taking the time to read our submission and offer constructive comments which we have used to substantially improve the paper.
>
> > The paper did a nice job of motivating then explaining the various models considered. However, I had a hard time following the theoretical analysis. Eg. I wish the authors had provided some details of the derivation of Eq1, or a pointer.
>
> We will provide a reference (Cox and Oakes, 1984) for the censored likelihood function in Eq (1) and explain it in details before Section A in the supplement.
>
>  D. R. Cox and D. Oakes. Analysis of survival data. CRC Press, 1984.
>
> > I also wondered which parts of Sec 3.2 were general for all neural nets, and what was specific to survival functions, and what just for variants of nn extensions of EH models?
>
> In Section 3.2, The description of neural networks and Holder class is general for the deep learning, but assumption (A1)-(A4) is specific to survival analysis.
>
> > Some parts could have been a bit more precise--e.g I assume that $\hat{h}$ is based on the sample size $n$ (line 225).
>
> We will make this  more precise by  emphasizing that the estimator $\hat{h}$ is based on the sample size $n$ in the comment right after Theorem 1.
>
> > Also, I did not understand how the theoretical analysis informed the development of the algorithm, or was used in the analysis of the empirical results.
>
> We know that deep learning has achieved tremendous success in practice, but the theoretical understanding still lags behind, at least for censored survival data.  We hope this paper, to some extent, is able to shed light on deep learning theories and guide the empirical studies. We have shown in Theorem 2 that the estimators attain fast convergence rates when the hyperparameters such as the number of hidden layers and neurons satisfy Assumption (A5) in paper. This helps practitioner to pre-set the hyperparameters before implementing algorithm. For example, in the experiments section, we select the search (candidate) space of the number of hidden layers to be small integers based on its number of order O(log n) in Assumption (A5), because log n is typically very small.
>
> > They empirically explored many (10) survival models, over 4 different datasets. The authors should have performed some statistical tests, to indicate which of the results were significant.
>
> This is a great suggestion as it will be very useful in practise to have a statistical significance test.  Unfortunately, the supporting theory is still lagging for deep survival models as it involves testing a nonparametric component. We made the first step to establish the convergence rate of the estimators but there is much remain to to be done before we can present a rigorous test. We will continue to explore this important open problem but want to emphasize that we have taken the important first step by providing the convergence rate of the estimators. The  importance of understanding the theory of deep learning is underscored by your question and what has been accomplished in this paper. This paper focuses on modelling DeepEH as a remedy for the proportional hazard assumption, when it is violated. We further provide theoretical support for the proposed model. Short of a rigorous statistical test for the adequacy of the proportional hazards assumption, we suggest to use some diagnostic tools, such as checking whether the log cumulative hazard functions (i.e., log$\Lambda(t|X)=$log$\Lambda_0(t) + g_0(X)$) of subjects are parallel. Typically, crossing log cumulative hazard functions suggests a violation of the proportional hazards assumption. We will add a brief discussion in the final version or supplement.
>
> > Given the paper’s concern with high dimensional features, it was surprising that the datasets were all low-dimensional. I wish the authors had empirically explored this issue, by using high dimensional data.
>
> This raises an interesting point about the curse of dimensionality, as it has different meaning in different contexts.  In this paper, the curse of high dimension is in the context of classical smoothing methods, such as kernel smoothing and B-spline, to estimate the nonparametric functions $h_1$ and $h_2$, whose implementations are often inefficient if there are more than four features. So a data set with more than four features will suffer the curse of dimensionality already if a nonparametric smoothing method is employed. In contrast, deep learning can handle four or higher dimensional comfortably and efficiently. The data sets in Section 4.3  have 5 to 12  features, which will pose substantial challenges to  nonparametric smoothing methods but not for the deep learning approach.
>
> This challenge of dimensionality commonly faced in survival analysis is very different from the high dimensional settings faced with e.g., genetic data, where the number of features (genes) exceeds the sample size.
>
> Before the deep learning era, researchers typically assumed low dimensional structure of the survival data, such as, nonparametric additive model (Stone, 1985) and single index model (Ichimura, 1993), to alleviate the curse of dimensionality. However, such low dimensional assumptions are very restrictive and may lead to model misspecification and low prediction power. Moreover, even if such a low-dimensional structure exists, DeepEH can detect it. This is where the deep learning approach really shines, as it can automatically learn to exploit low dimensional structure in the underlying model.
>
> We will precisely clarify this confusion around "high" dimensionality in the revised paper.
>
> H. Ichimura. Semiparametric least squares (SLS) and weighted SLS estimation of single-index models. Journal of Econometrics, 58: 71--120, 1993.
>
> C. J. Stone. Additive regression and other nonparametric models. Annals of Statistics, 13: 689--705, 1985.
>
> > While IBS is considered both discriminative and calibration, it would have been useful to see if each dataset was calibrated--perhaps at the median time. See the analysis in https://jmlr.csail.mit.edu/papers/v21/18-772.html .
>
> From Table 1, we note that the proposed DeepEH perform better than other nine methods in terms of C-index (discrimination) and IBS (discrimination and calibration) and the IBS obtained from DeepEH is very small compared to the  other methods. It is known that evaluation based on a single value of the survival distribution, such as median survival time, is fairly unreliable  (Henderson and Keiding, 2005). That said, we are working to have a discussion on 1-calibration or D-calibration (Haider et al, 2020) for the proposed model in the revision.
>
> R. Henderson and N. Keiding. Individual survival time prediction using statistical models. Journal of Medical Ethics, 31: 703-706, 2005.
>
> H. Haider, B. Hoehn, S. Davis, R. Greiner. Effective ways to build and evaluate individual survival distributions. Journal of Machine Learning Research, 2020.
>
> > wrt the IBS: Li258, \hat{G} is NOT the KM estimator. It is the estimator of the conditional survival function of the censoring times calculated using the Kaplan-Meier method. Is the integral Eq12 done analytically for your models, or estimated empirically? If the latter, are there issues related to the intervals considered? Is $\tau$ the largest time (censored or not), or the largest uncensored time, or something else?
>
> Thanks for pointing this out. In the revision, we will clarify  \hat{G}(t) as "the estimator of G(t):=P(C>t) for the censoring time $C$ based on the Kaplan-Meier method."
>
> The IBS in Eq. 12 is calculated through the Riemann sum of Brier score BS(t) evaluated at equal grid on $[0,\tau]$. Here $\tau$ is the largest observed time (censored and uncensored time). We will  clarify it in the paper.
>
> > The authors should have discussed how they split the data into 5 folds–was this balanced for (censored, death) times? For anything else?
>
> We will clarify that ``each data is randomly split into five equal folds based on their subject ID." Such a split should produce a good balance of censored times since it is randomly selected.
>
> > I wish they plotted the results in a figure–perhaps moving Table 1 to the appendix. Similarly, it would be helpful if they had plotted the hazard curves, or the survival curves, to help the reader visualize the results.
>
> We will move Table 1 to the supplement and replace it in the main text with a figure that is visually more appealing.
>
> > li55: given the many different ways to evaluate a survival model, what does “improvement” mean? li77: I assume “omitted features” means some values are missing for some instances. If so, are these omissions considered MCAR?
>
> We will revise line 55 to "...,but this extension failed to provide reliable improvements in terms of concordance index [56]".
>
> In line 77,  ``omitted features" means these features were not observed at all for all subjects, hence they are not  missing data.
>
> To avoid confusion, we will replace "omitted features'' by  "unobserved features'' in  line 77 with sentence "relative to CoxPH, AFT models are more stable when accounting for omitted features."
>
> > li277: Why mention competing risks? Isn’t there only a single type of risk?
>
> In line 277, we are referencing the DSM method of Nagpal et al. (2021) which was originally proposed for competing risks although it can also model single event survival data. We will revise the statement "DSM developed a fully parametric method to estimate the competing risks for survival time [41]." to ``DSM developed a deep learning framework to estimate the survival function that can also handle competing risks in addition to single event survival data [41]."
>
> C. Nagpal, X. R. Li, and A. Dubrawski. Deep survival machines: Fully parametric survival regression and representation learning for censored data with competing risks. IEEE Journal of Biomedical and Health Informatics, 2021.
>
> > li119: extra $\beta$
>
> Thanks for point the typo in line 119 which we've fixed.

---

> ### Comment · Reviewer_hDJX · 2021-08-28
> **Feedback on Authors' Feedback**
>
> Thanks for your feedback, which clarified some issues, but not all.
>
> It is still not clear how much the theory informed practice, nor how the empirical results confirmed the theoretical claims.
>
> wrt statistical tests: I was not asking about violations of the proportional hazard assumption, but was curious to see which of the models was (statistically) best, and then to understand why--to see if these results confirm the theoretical claims.
>
> wrt dimensionality: Does this mean the model is unable to deal with more than a dozen features?
>
> Note there are of course many survival models that do not require the proportional hazard assumption, and that can accommodate many more than 12 features.
>
> Bottom line:  The feedback helped some, but still left many issues.  I am therefore leaving my evaluation at 6.

---

> > ### Author Response · Authors · 2021-08-31
> > **Response to Reviewer hDJX**
> >
> > We thank you for your comments and provide our response below.
> >
> > >It is still not clear how much the theory informed practice, nor how the empirical results confirmed the theoretical claims.
> >
> > We have addressed this partially by pointing out that the theory led us to select networks with only a few hidden layers. For example, in the experiments section, we select the search (candidate) space of the number of hidden layers to be small integers based on its number of order $O(\log n)$ in Assumption (A5). The empirical results in Section C of the Appendix also do support our theory.  Moreover,  Theorem 2 confirms that DeepEH  is able to detect the low-dimensional structure of survival data as reflected in the convergence rate of the nonparametric estimates of $h_1$ and $h_2$. This type of theory provides useful understanding of conditions under which the proposed DeepEH approach is guaranteed to work well.
> >
> > >wrt statistical tests: I was not asking about violations of the proportional hazard assumption, but was curious to see which of the models was (statistically) best, and then to understand why--to see if these results confirm the theoretical claims.
> >
> > To see which model fits the data best, one could use the C-index or integrated Brier score to determine that and we have done so in the data analysis. For the four data sets we have explored,  the  DeepEH model generally provided the best fit.
> > If we want to test  whether the finding is statistically significant, we can test  $\mathcal{H}_0: g_1=0, g_2=0$ v.s. $\mathcal{H}_1: g_1\neq0 \  \text{or} \ g_1\neq0.$  Such a tests would require, as a first step, estimators of $g_1$ and $g_2$  under $\mathcal{H}_0$ and the DeepEH model, respectively. Then one could apply the nonparametric likelihood ratio test to carry it out.   Alternatively, if the question is whether the DeepEH model is significantly better than the DeepCox model, the hypothesis would be $\mathcal{H}_0: g_1=0$ v.s. $\mathcal{H}_1: g_1\neq0 $. Again, this requires the estimation of $g_1$ and $g_2$ and a nonparmatric likelihood ratio test could be applied subsequently.
> >
> > Our analysis in the paper provides the  theoretical foundation for this test through the rate of convergence established in Theorem 2. But to actually carry out a nonparametric likelihood ratio test is no small feat as we explained before.  This will be an interesting future project.
> > We emphasize here that no one has come up with such a test for survival data based on any deep learning approach. The theory we develop in this paper provides  a first step towards this direction.
> >
> > > wrt dimensionality: Does this mean the model is unable to deal with more than a dozen features?
> > Note there are of course many survival models that do not require the proportional hazard assumption, and that can accommodate many more than 12 features.
> >
> > The model itself has no limit on the number of features, it can handle as many as any other survival models. The issue with high-dimensional features is in the implementation of the algorithm, which will require some form of regularization, if the sample size is not large enough to provide reliable estimates under many features. In this paper, we are already confronting the first question - what does high-dimensional features mean? How large does the sample size need to be before substantial regularization should be applied? Our theory reveals the answer will depend on the intrinsic dimension of the model, so the answer does not solely depend on the number of features.
> >
> > One standard regularization method is to add a penalty (e.g. $L_1$ or SCAD) to the loss function to penalize the number of features over which the learned function varies.  However, how to do this optimally requires extensive exploration, which we leave for future work. Note there is a huge body of recent research just on feature selection for standard regression with neural networks, indicating that good methods are far from established even in this well-studied setting, whereas we are here dealing with neural networks for censored survival data, for which new studies will be needed.

---

> > > ### Comment · Reviewer_hDJX · 2021-09-04
> > > **Response and Restatements**
> > >
> > > Thank you for your response.  Of course, even without that theoretic analysis, most researchers would probably have used a network with only a few hidden layers. Also, this use of O( log n) was an assumption, not the result of the analysis.  While I realize the theorem claims that this is {\bf sufficient} for the conclusion, it would have greater practical importance if one could show that this assumption was important, by showing what happened if this assumption was violated.   If not theoretically, perhaps showing empirically that models that violate this do poorly, with statistically inferior concordance or IBS.
> > >
> > > While Table 1 does provide many empirical results, it does not state which models were *statistically significantly better*.
> > >
> > > It would have been interesting if the empirical studies included models with many features, to demonstrate how this differs from “intrinsic dimension”.

---

### Official Review · Reviewer_cez1 · 2021-07-17

**Rating:** 6
**Confidence:** 4

**Summary:**

The authors propose to extend the "Extended Hazards Model" (EH) with deep learning approaches to model right censored time to event data. EH model is a generalization of both the Accelerated Failure Time and the Cox Model Proportional Hazards Model and thus is more flexible than either of the two approaches. They demonstrate empirically that the extended hazards model improves the prediction of survival across multiple datasets in terms of both discriminative performance (C-Index) and Integrated Brier Score on multiple competitive baselines.



**Limitations And Societal Impact:**

The authors should clearly describe how strong assumptions on the hazard distributions can lead to miss estimation of the risk profile/ trajectory for individuals leading to potentially harmful decision making.

**Main Review:**


Learning survival models to model right censored datasets with neural networks is a challenging problem. The authors propose to employ the extended hazards model to model survival data. Their motivation is well justified intuitively. The EH is a generalization of both the Cox Proportional Hazards model, and the Accelerated Failure Time model and thus alleviates challenges associated with these two models.

## Strengths

- Rigorous experimental evaluation across multiple strong datasets and baselines, in terms of both Discriminative Performance and Integrated Brier Score.
- A major challenge with the CoxPH model is its limitations in handling problems where the PH assumptions are not met. The Extended Hazards approach is well justified intuitively since, it can alleviate these challenges.
- Some interesting theoretical justification for the use of Deep Learning to learn low dimensional representation to better estimate the time-to-event distribution.

## Weaknesses:

- Although the paper attempts at providing a theoretical justification, in Sec 3.2. The theoretical results are largely a consequence of the convergence rates and results already established in (Y.-K. Tseng and K.-N. Shu., 2011) and (Zeng and Lin, 2007) making this part of the contribution, much less novel.
- I would thus evaluate this paper as a modeling paper, and not necessary as theoretical contribution. Unfortunately, in terms of modelling the main text of the paper glosses over many important details of the Extended Hazards model which would be required for reproducibility by a reader not deeply entrenched in survival analysis. For example,

A) it is unclear how Eq. (4) -> Eq. (5), since the authors don't explicitly mention the relationship between the hazard function and the survival function.

B) In the estimate of the Pseudo-Likelihood (post line 165),  it is unclear how the Kernel $\mathcal{K}(\cdot)$ shows up as it has not been referred to before. In any case, the likelihood $\mathcal{L}(\cdot)$ should be explicitly rewritten as a function of the Kernel $\mathcal{K}_b(\cdot)$ with bandwidth $b$.

C) It is unclear as to how $\Phi(\cdot)$ is computed. (It is an integral over event times) is this computed over a fixed discrete grid or over all observed event times?

- Finally, I have major concerns about reproducibility of this paper. For example, in the main text and the supplement, the authors do not mention the choice of the  kernel function ($\mathcal{K}(\cdot)$) they use or how the bandwidth is selected. While the number of baselines considered are extensive; there is no discussion about the hyperparameter grid over which the baselines were tuned.

Overall, the paper proposes an interesting extension to the original idea of extended hazards using Deep Learning, which is welcome. The paper also aims to provide theoretical justification for their choice, however some of the implementation and design details are ignored in the paper calling into question the reproducibility of the proposed approach.


## Other Concerns/Recommendations:

- Have the authors evaluated performance in terms of Concordance over truncated event times? If yes, they are encouraged to include these results in the supplement to establish how the discriminative performance varies at different time horizons.
- An important problem that the model aims to alleviate is the strong assumption of proportional hazards. The authors are recommended to include plots of the estimated survival distributions on real world data in the main text to demonstrate the model is able to capture scenarios where the assumption is violated. (One clear situation of this is when the estimated survival curves of two individuals intersect)
- The authors are recommended to keep the theorems in the main text but to move exact details of Section 3.2 to the supplement, and use the extra space to address the concerns raised above.
- Have the authors compared the performance of their approach to stratified Cox/AFT approaches (separate cox/aft models for different strata) or approaches that involve learning mixtures of Cox/aft models (Discrete mixture regression models for heterogenous time-to-event data: Cox assisted clustering, Kevin H. Eng & Bret M. Hanlon, 2012; Deep Cox Mixtures for Survival Regression, Nagpal et al, 2021; )? If not, can the authors comment when should an EH model be preferred over using stratified Cox models?

**Time Spent Reviewing:**

3

---

> ### Author Response · Authors · 2021-08-10
> **Response to Reviewer cez1**
>
> We appreciate your comments, which helped us to improve the paper accordingly.
>
> > Although the paper attempts at providing a theoretical justification, in Sec 3.2. The theoretical results are largely a consequence of the convergence rates and results already established in (Y.-K. Tseng and K.-N. Shu., 2011) and (Zeng and Lin, 2007) making this part of the contribution, much less novel.
>
> Although DeepEH is an extension of  Tseng and  Shu (2011) and Zeng and Lin (2007), the theoretical approaches and results are substantially different from theirs. First, they modelled the effect of the features through  a single index, i.e., a linear combination of features, which is a parametric approach. So their proofs mainly involve  Taylor expansions to derive the asymptotic theory of the estimators. In contrast, we model the effects of the features nonparametrically through neural networks, and we approach  the convergence rate of the estimators through the  nonparametric  M-estimation framework, which requires comprehensive knowledge of empirical process and deep neural networks.
>
> Moreover, a  main goal of the paper is to provide a theoretical understanding of the deep learning application for survival data. We have shown that, by identifying the low-dimensional structure of the data, deep learning estimators share faster convergence rates than classical nonparametric approaches. This profound nonparametric theoretical result distinguishes our contribution from parametric results by Tseng and  Shu (2011) and Zeng and Lin (2007).
>
>
> > I would thus evaluate this paper as a modeling paper, and not necessary as theoretical contribution.
>
> Thanks for the acknowledgement of our practical modeling contribution. It should be noted that we not only aim to propose a new model that is better than the state-of-art deep learning survival models, but also provided theoretical support for the use of deep learning in such a model. To our knowledge, very little is known mathematically about deep survival modeling. Thus our theoretical contribution is an added benefit to the literature, especially because it sheds light on practice. For example, we select the search (candidate) space of the number of hidden layers to be small integers based on the optimal number of hidden layers (of order O(\log n)) in Theorem 2 and Assumption (A5). Hence, the contribution of our paper is not just limited to modeling.  We believe this paper will also help establish theoretical understanding for other deep learning applications to survival analysis.
>
> > Unfortunately, in terms of modelling the main text of the paper glosses over many important details of the Extended Hazards model which would be required for reproducibility by a reader not deeply entrenched in survival analysis. For example, it is unclear how Eq. (4) $\rightarrow$ Eq. (5), since the authors don't explicitly mention the relationship between the hazard function and the survival function.
>
> We have rewritten the exposition in our modeling section to more precisely describe the components of the EH model. In our revision, we've clarified the derivation from Eq. (4) to Eq. (5) as follows:
>  By the definition of conditional survival function  $S(t|X)=e^{-\int_0^t\lambda(s|X)ds}$, we have $S(t|X)=e^{-\int_0^t\lambda(s|X)ds}=e^{-\int_0^t\lambda_0(s e^{h_1(X)})e^{h_2(X)}ds}=e^{-\Lambda_0(te^{h_1(X)})e^{h_2(X)-h_1(X)}}.$
>
> > In the estimate of the Pseudo-Likelihood (post line 165), it is unclear how the Kernel $K(\cdot)$ shows up as it has not been referred to before. In any case, the likelihood $L(\cdot)$ should be explicitly rewritten as a function of the Kernel $K_b(\cdot)$ with bandwidth b.
>
> In line 166, we will explain it with ``where $K(\cdot)$ is a known kernel function satisfying Assumption (A4) below. In Section 4, we adopt the Gaussian kernel."
> We would like to keep the notation $\mathcal{L}_n(h_1,h_2)$ of log pseudo-likelihood to be in line with  the previous works by Tseng and  Shu (2011) and Zeng and Lin (2007).
>
> > It is unclear as to how $\Phi(\cdot)$ is computed. (It is an integral over event times) is this computed over a fixed discrete grid or over all observed event times?
>
> We will clarify in the experiments section that:
> We use a Gaussian kernel as our kernel function $K(\cdot),$ so the function $\Phi(t)=\int_0^{t}K(u)du$ is simply the cumulative distribution function of the standard normal distribution.
>
>
> > in the main text and the supplement, the authors do not mention the choice of the kernel function $K(\cdot)$ they use or how the bandwidth is selected. While the number of baselines considered are extensive; there is no discussion about the hyperparameter grid over which the baselines were tuned.
>
> We will clarify that: As previously stated, we use a Gaussian kernel. The optimal order of the  kernel bandwidth is $O((8\sqrt{2}/3)^{1/5} n^{-1/5})$ for kernel smoothing methods (Jones 1990; Jones and Sheather 1991). In our experiments, we merely pre-select the bandwidth to be $(8\sqrt{2}/3)^{1/5} n^{-1/5}$. The results of baseline methods are obtained by running the code  provided in the corresponding papers, including the choice of hyperparameters, which will be listed in full detail in our revised Appendix.
>
> > An important problem that the model aims to alleviate is the strong assumption of proportional hazards. The authors are recommended to include plots of the estimated survival distributions on real world data in the main text to demonstrate the model is able to capture scenarios where the assumption is violated. (One clear situation of this is when the estimated survival curves of two individuals intersect). The authors are recommended to keep the theorems in the main text but to move exact details of Section 3.2 to the supplement, and use the extra space to address the concerns raised above.
>
> Thanks for the great suggestion. The violation of the proportional hazards model can easily be demonstrated graphically by showing crossing survival functions based on different feature values. In our revised paper, we will plot several randomly selected survival curves in the paper and relegate the description of the neural networks and H{\"o}lder class to the supplement.
>
>
> > Have the authors compared the performance of their approach to stratified Cox/AFT approaches (separate Cox/AFT models for different strata) or approaches that involve learning mixtures of Cox/AFT models (Discrete mixture regression models for heterogenous time-to-event data: Cox assisted clustering, Kevin H. Eng \& Bret M. Hanlon, 2012; Deep Cox Mixtures for Survival Regression, Nagpal et al, 2021; )? If not, can the authors comment when should an EH model be preferred over using stratified Cox models?
>
> We have not compared the performance of our approach with stratified Cox/AFT for several reasons. Although stratified Cox models are a useful extension of the standard Cox models to allow for features with non-proportional hazards,  the stratified Cox models still require the proportional hazards assumption at each stratum and it is not clear how one could pursue such stratum.  The EH model offers an alternative to the stratified Cox models when the proportional assumption is violated at some stratum. The aforementioned diagnostic plot can be used to visually inspect the proportional hazards assumption. We will add a brief discussion in the paper if there is space for it.
>
>
> > Have the authors evaluated performance in terms of Concordance over truncated event times? If yes, they are encouraged to include these results in the supplement to establish how the discriminative performance varies at different time horizons.
>
> It is actually difficult to calculate concordance index for arbitrary  truncated event times since they are censored and thus unknown. This is in fact a weakness  of the C-index as it cannot record the concordance of paired censored event times. This is why we also implemented the integrated Brier score, which involves both censored and uncensored times, to get a more complete evaluation.

---

### Official Review · Reviewer_kQKv · 2021-07-19

**Rating:** 7
**Confidence:** 3

**Summary:**

This work proposes DeepEH, a survival analysis framework that formulates the hazard function as \lambda(t|X) = \lambda_0(t.exp(h_1(x))).exp(h_2(x)). This formulation has the Cox PH and AFT as special cases. The authors estimate h_1 and h_2 by neural networks and provide theoretical support for the consistency and convergence rate of the resulting survival function estimator.

**Limitations And Societal Impact:**

Yes

**Main Review:**

1) I saw a few errors in the English and a few typos: "adopt a parametric approaches", "are asymptotically consistent 90 and enjoys fast convergence rates", "advantageousß", "aasumes", "The data sets consists of"

2) What do you mean by "omitted features" in "Keiding et al. [33] also show that, relative to CoxPH, AFT models are more stable when accounting for omitted features."?

3) When you say "Specially, the convergence rates are determined by the intrinsic dimension of the underlying functions rather than the original high-dimension input features." and then go on to say "This, to some extent, explains the great success of deep learning and distinguishes our contributions from existing work.", how is your modeling different from previous survival analysis frameworks that make use of deep learning? In other words, what sets your work apart from other survival analysis algorithms that use deep learning given they are using neural networks too?

4) In section 2.1, when you say "It is difficult to estimate and directly from the likelihood function", it would have been beneficial to add a sentence or two explaining why.

5) When you say "This is the main reason why the Cox proportional hazards model dominates the field of survival analysis for nearly fifty years" what is the main reason of Cox's domination? Optimizing the parametric and non-parametric components separately?

6) In section 3.1, the pseudo-likelihood can be better justified. The intuition behind the formula of the loss is missing. It seems to me that the pseudo-likelihood and the hazard function that is introduced afterwards can switch places. A better storyline would have been: (according to my understanding of what this section is trying to say) Following [52] the hazard function can be approximated by the following formula because ... which would then give rise to a pseudo-likelihood (why is it "pseudo" in the first place?) which is ...

7) In section 4.1, the evaluation criteria can be intuitively more elaborated on. The C-index is a bit explained when you talk about assessing the concordance between rankings but the Integrated Brier Score is not explained much. Giving more intuition to the reader would go a long way in understanding how these criteria are useful in assessing survival estimators.

8) How are you recovering the survival function from the hazard function? Are you using numerical integration methods?

9) In section 4.2, why DeepHit only performs well when no lower dimensional structures exist for the survival times?

10) Table 1 could use rearrangement. It is difficult to read. Could be decluttered.

**Time Spent Reviewing:**

5

---

> ### Author Response · Authors · 2021-08-10
> **Response to Reviewer kQKv**
>
> Thank you for your insightful and valuable feedback and for your support of the paper.
>
> > When you say  ``Specially, the convergence rates are determined by the intrinsic dimension of the underlying functions rather than the original high-dimension input features.'' and then go on to say "This, to some extent, explains the great success of deep learning and distinguishes our contributions from existing work", how is your modeling different from previous survival analysis frameworks that make use of deep learning? In other words, what sets your work apart from other survival analysis algorithms that use deep learning given they are using neural networks too?
>
> Existing works of deep survival models mainly focus on the CoxPH framework, which  assumes  that the log hazard functions for individuals are parallel to each other. Such an assumption are for mathematical convenience and may not hold  in reality. Using the powerful representation to complex model of deep learning, we proposed the DeepEH to avoid model mis-specification. This distinguished our work from existing survival analysis algorithms as we offer a more flexible model.  That said, a more flexible model often pay a price of higher variability. Luckily, this is not the case with DeepEH as we establish theoretical support for our approach and show that,
>  even under the more complicated EH model, deep learning is able to detect the low-dimension structure of the data embedded in a higher-dimensional space. Specifically, we mean that although the number of features $p$, which is the dimension of $X$, may be high, the key information contained in these features may reside in a lower dimensional subspace of $X$, which is determined by the intrinsic dimension $\boldsymbol{p}^*$ (in Section 3.2) of the nonparametric functions $h_1$ and $h_2$.
>
> Our theory  (Theorem 2) suggests that deep learning  is able to detect this lower dimensional subspace and it is this intrinsic dimension $\boldsymbol{p}^*$ that determines the convergence rate of the estimators for $h_1$ and $h_2$. In contrast, had a nonparametric smoothing method been employed to estimate them, the convergence rate would be determined by the dimension $p$ of the data, which may be much slower than the convergence rate of a  deep learning estimate (cf. Theorem 2).
>
>
> > In section 2.1, when you say "It is difficult to estimate and directly from the likelihood function", it would have been beneficial to add a sentence or two explaining why.
>
> We will replace "It is difficult to estimate $\lambda_0$ and $\beta$ directly from the likelihood function (1)" by ``It is difficult to estimate $\lambda_0$ and $\beta$ directly from the likelihood function (1) because the optimization with respect to $\lambda_0$, a nonparametric function, would be challenging.   In fact, this likelihood is unbounded  due to  the unconstrained nature of the nonparametric function $\lambda_0$".
>
> > When you say "This is the main reason why the Cox proportional hazards model dominates the field of survival analysis for nearly fifty years" what is the main reason of Cox's domination? Optimizing the parametric and non-parametric components separately?
>
> Yes, because the partial likelihood does not involve the nonparametric component $\lambda_0$, the regression parameter $\beta_0$ in the CoxPH model can be estimated without involving the nonparametric component. Once a parametric estimate of $\beta_0$ has been obtained, the baseline hazard function $\lambda_0$ can be estimated easily from the method of Breslow (1972).
>
> We will replace "This is the main reason why the Cox proportional hazards model dominates the field of survival analysis for nearly fifty years"  by ``This attractive feature that the simple parameter $\beta_0$ can be estimated directly through the partial likelihood  without requiring nonparametric estimation of $\lambda_0$ is the main reason why the Cox proportional hazards model has dominated the field of survival analysis for nearly fifty years".
>
> N. E. Breslow. Discussion of the paper by DR Cox. Journal of the Royal Statistical Society Series B (Statistical Methodology), 34(2): 216--217,1972.
>
> > In section 3.1, the pseudo-likelihood can be better justified. The intuition behind the formula of the loss is missing. It seems to me that the pseudo-likelihood and the hazard function that is introduced afterwards can switch places. A better storyline would have been: (according to my understanding of what this section is trying to say) Following [52] the hazard function can be approximated by the following formula because ... which would then give rise to a pseudo-likelihood (why is it ``pseudo" in the first place?)
>
> As suggested, we will revise the statement clarifying that:
> "Since the log-likelihood is unbounded, direct maximization of the likelihood function will not work. Instead, a pseudo likelihood, as suggested in  [52], will be employed instead.  Specifically, we first use the kernel smoothing method to approximate the hazard function with fixed $h_1$ and $h_2$ and then plug the the smoothed hazard function into the log-likelihood function (1) yielding the following pseudo-likelihood function for the DeepEH model: ..."
>
> > In section 4.1, the evaluation criteria can be intuitively more elaborated on. The C-index is a bit explained when you talk about assessing the concordance between rankings but the Integrated Brier Score is not explained much. Giving more intuition to the reader would go a long way in understanding how these criteria are useful in assessing survival estimators.
>
> Thanks for the suggestion. We will add  the following statement about Integrated Brier score. `` Similar to the mean squared error, the Brier score (Brier, 1950)  is a measure of performance based on the predicted probability for binary data.  Graf et al. [20] extended the Brier score to right censored data in order to assess the accuracy of an estimated survival function at some time t."
>
> G. W. Brier. Verification of forecasts expressed in terms of probabilities. Monthly Weather Review 78: 1–3, 1950.
>
>
> > How are you recovering the survival function from the hazard function? Are you using numerical integration methods?
>
>  Yes, we used Riemann sum of the baseline hazard function to get the cumulative baseline hazard function, which then leads to the survival function. We will clarify this at the end of Section 3.1 and mention  ``In Section 4, we employ Riemann sum of the estimated baseline hazard function to estimate the cumulative baseline hazard function, which then lead to an estimate of the survival function as $S(t|X)=e^{-\\Lambda(t|X) }$."
>
>
> > In section 4.2, why DeepHit only performs well when no lower dimensional structures exist for the survival times?
>
> DeepHit is a fully nonparametric model to estimate the survival function, which will be disadvantaged by a  parametric or semiparametric model if the data come from such  models. For example, when the true model is CoxPH, whose linear risk predictor leads to  a one-dimension data structure,  but DeepHit would treat it as a $p$-dimensional nonparametric estimation problem.  Thus, the CoxPH method is expected to  outperform DeepHit.
>
>
> > Table 1 could use rearrangement. It is difficult to read. Could be decluttered.
>
> We will move Table 1 to the supplement and replace it in the main text with a figure that is visually more appealing.
>
> > What do you mean by "omitted features" in ``Keiding et al. [33] also show that, relative to CoxPH, AFT models are more stable when accounting for omitted features."?
>
> The "omitted features" are  "unobserved features". We will replace "omitted features'' with ``unobserved features".
>
> > I saw a few errors in the English and a few typos: "adopt a parametric approaches", "are asymptotically consistent 90 and enjoys fast convergence rates", "advantageousß", "aasumes", “The data sets consists of".
>
> Thanks for pointing these out. We have corrected them and made many other language improvements in the revised version of this paper.

---

### Decision · Program_Chairs · 2021-09-27

**Decision:**

Accept (Poster)

**Comment:**

This paper has been reviewed by four knowledgeable referees resulting in one accept, two marginal accept and one marginal reject recommendations. Most of the criticism centers on limited novelty and utility of the theoretical portion of this work. Yet, the proposed method appears to stand on its own, yet it has been evaluated on a relatively low dimensional data only. The authors have engaged in discussions with the reviewers that helped resolve some but not all of the stated concerns. All things considered, I recommend this paper for acceptance if space permits its inclusion in the conference agenda. The authors should strive to incorporate all constructive recommendations from the reviewers in the finał camera ready version of their paper.